# DEEM : Diffusion Models Serve as the EyEs of Large Language Models for Image Perception

**Run Luo**[1,2*] **Yunshui Li**[1,2*] **Longze Chen**[1,2*] **Wanwei He**[1,2] **Ting-En Lin**[3]
**Ziqiang Liu**[1,2] **Lei Zhang**[1,2] **Zikai Song**[4] **Hamid Alinejad-Rokny**[5]
**Xiaobo Xia**[6,7] **Tongliang Liu**[8] **Binyuan Hui**[9†] **Min Yang**[1†]

[1]Shenzhen Key Laboratory for High Performance Data Mining, SIAT, CAS
[2]University of Chinese Academy of Sciences      [3]Tsinghua University
[4]Huazhong University of Science and Technology [5] University of New South Wales
[6] School of Computing, National University of Singapore
[7] MoE Key Laboratory of Brain-inspired Intelligent Perception and Cognition,
University of Science and Technology of China [8] The University of Sydney [9]Alibaba Group

## Abstract

The development of large language models (LLMs) has significantly advanced the emergence of large multimodal models (LMMs). While LMMs have achieved tremendous success by promoting the synergy between multimodal comprehension and creation, they often face challenges when confronted with out-of-distribution data, such as which can hardly distinguish orientation, quantity, color, structure, etc. This is primarily due to their reliance on image encoders trained to encode images into task-relevant features, which may lead them to disregard irrelevant details. Delving into the modeling capabilities of diffusion models for images naturally prompts the question: Can diffusion models serve as the eyes of large language models for image perception? In this paper, we propose DEEM , a simple but effective approach that utilizes the generative feedback of diffusion models to align the semantic distributions of the image encoder. This addresses the drawbacks of previous methods that solely relied on image encoders like CLIP-ViT, thereby enhancing the model's resilience against out-of-distribution samples and reducing visual hallucinations. Importantly, this is achieved without requiring additional training modules and with fewer training parameters. We extensively evaluated DEEM on both our newly constructed RobustVQA benchmark and other well-known benchmarks, POPE and MMVP, for visual hallucination and perception. In particular, DEEM improves LMM's visual perception performance to a large extent (e.g., 4% ↑ on RobustVQA, 6.5% ↑ on MMVP, and 12.8 % ↑ on POPE ). Compared to the state-of-the-art interleaved content generation models, DEEM exhibits enhanced robustness and a superior capacity to alleviate model hallucinations while utilizing fewer trainable parameters, less pre-training data (10%), and a smaller base model size. Extensive experiments demonstrate that DEEM enhances the performance of LMMs on various downstream tasks without inferior performance in the long term, including visual question answering, image captioning, and text-conditioned image synthesis. The code and benchmark are available at https://github.com/RainBowLuoCS/DEEM

## 1 Introduction

With the success of large language models (LLMs), large multimodal models (LMMs) built on LLMs have garnered significant attention. Researchers (Liu et al., 2024a; Zhu et al., 2023; Dai et al., 2024; Alayrac et al., 2022; Chen et al., 2023) have attempted to build a bridge between large language models and image encoders through simple mapping modules, and have already made significant progress in multimodal understanding tasks such as visual question answering. Subsequent

---

*Equal contribution. † Min Yang and Binyuan Hui are corresponding authors.

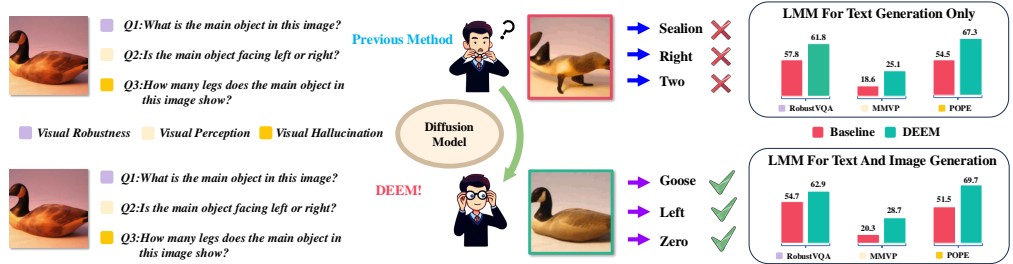

Figure 1: **Illustration of our DEEM .** When encountering natural adversarial examples or out-of-distribution data, DEEM uses the diffusion model to check if the semantic features of the image encoder match the input images. This approach allows DEEM to serve as the "eyes" of the large language model, proactively identifying and correcting misinterpreted semantic information during training, thereby avoiding the loss of important visual details. This enhances the robustness, hallucination recognition, and foundational visual perception capabilities of LMMs. In contrast, other models rely too heavily on erroneous inputs from the image encoder, making it difficult for them to handle challenges posed by such data.

studies (Yu et al., 2023; Sun et al., 2023b; Dong et al., 2023; Tian et al., 2024) utilize extra advanced diffusion models (DMs) (Rombach et al., 2022) for image generation and train the LMMs on interleaved text-image data in an end-to-end manner. This unified paradigm of multimodal understanding and creation brings various isolated multimodal tasks together, greatly boosting model capabilities and expanding application scenarios.

However, these models commonly rely on encoder architectures like CLIP-ViT (Radford et al., 2021), which suffers from certain perceptual understanding limitations due to the contrastive learning paradigm and the noisy image-text pairs used in training, to encode input images. Additionally, these image encoders are typically trained to encode images into features relevant to downstream tasks, thereby disregarding irrelevant details. Consequently, as shown in Fig. 1, when faced with images outside the training scope, they often capture biased semantic features, resulting in erroneous visual information being perceived by subsequent language models. This accumulation of inaccuracies renders the multimodal model unable to comprehend multimodal context effectively. For this reason, this makes it difficult for previous methods to discern subtle details, thereby hindering their ability to handle tasks related to basic visual perception, visual hallucinations, and visual robustness that are very simple for humans.

On the contrary, the goal of diffusion models (Ho et al., 2020a) is to learn a diffusion process that characterizes a probability distribution for a given dataset, without direct training on the downstream task objective. This enables it to capture finer details of images for better handling of out-of-distribution data. However, there have been few efforts to integrate the capabilities of the diffusion model into the image perception of large multimodal models.

In this paper, we propose **DEEM**, a simple but effective approach to leverage the generative feedback of diffusion models for aligning the semantic distributions of image encoders in an elegant self-supervised manner. Building upon this, we introduce an end-to-end interleaved image-text generative modeling approach, where diffusion models serve as additional eyes of large language models for image perception. This addresses the limitations of previous methods that solely relied on image encoders such as CLIP-ViT (Radford et al., 2021), enhancing the model's robustness against out-of-distribution samples and reducing hallucination perception in multimodal scenarios, without the need for additional training modules and with fewer training parameters. To the best of our knowledge, we are the first to apply diffusion models to large multimodal models for image perception.

Specifically, DEEM takes interleaved image-text pairs as input to the model. It starts by encoding images and text using corresponding visual and text encoders, resulting in image tokens and text tokens. These tokens are then organized according to their original layout and inputted into a large language model to generate corresponding hidden state outputs. The model employs autoregressive modeling for the hidden state outputs of text and utilizes the output hidden states of images, along

with the image tokens encoded by the image encoder, as diffusion conditions. These conditions are then fed into a diffusion model for image reconstruction. Through end-to-end training, the model not only acquires the capacity to generate text and images but also employs semantic consistency regularization on the semantic information produced by the image encoder during image reconstruction. This compels the image encoder to incorporate more details into the semantic representation of the image, thereby mitigating the issue of semantic bias in image encoding.

DEEM is trained on a mixture corpora of image-text pairs and interleaved image-text sequences data without extra in-house data following previous solution (Li et al., 2022; 2023a; Dong et al., 2023; Tian et al., 2024). To assess the robustness recognition capability of LMMs, we constructed a new robustness benchmark, RobustVQA, based on existing datasets containing natural adversarial samples and out-of-distribution data. RobustVQA is divided into three parts: RobustVQA-A, RobustVQA-R, and RobustVQA-V, based on different data sources, aiming to provide better insights into the performance of LMMs in real-world scenarios. We conducted extensive evaluations of DEEM on RobustVQA and two widely recognized benchmarks, POPE and MMVP, for visual hallucination and perception respectively. Experimental results indicate that our method exhibits enhanced robustness, a superior capacity to alleviate model hallucinations and better visual perception ability in comparison to the state-of-the-art interleaved image-text modeling model MM-Interleaved (Tian et al., 2024), using a smaller-scale image encoder (CLIP-ConvNext-B (Liu et al., 2022) vs. CLIP-ViT-L (Radford et al., 2021)), a smaller-scale language model (Vicuna 7B vs. Vicuna 13B (Zheng et al., 2024)), and less pre-training data (without Laion-coco (Andreas et al., 2022) & Laion-en (Schuhmann et al., 2022)). DEEM outperforms MM-Interleaved 9.4% on RobustVQA, 17.8% on POPE and 9.1% on MMVP. Moreover, with further enhancement via supervised fine-tuning, DEEM achieves competitive results on various multimodal tasks, including visual question-answering, region-level image captioning, and text-to-image generation.

Before delving into details, we summarize our contributions as follows.

• **Robustness Benchmark.** We design a new robustness benchmark RobustVQA for LMMs based on publicly available ImageNet-A (Hendrycks et al., 2021b), ImageNet-R (Hendrycks et al., 2021a), and ImageNet-V2 (Recht et al., 2019) datasets, which can be utilized to effectively assess the visual robustness capabilities of the multimodal models.

• **Effective Method.** We are the first to introduce the diffusion model into the image perception of large language models, to correct potential semantic bias in the image encoder and alleviate the excessive compression of visual details. This approach enhances the model's robustness and hallucination mitigation capabilities without the need for additional modules or trainable parameters.

• **DEEM Model.** Based on the proposed method, we train a multimodal model with end-to-end interleaved text-image modeling capabilities. After supervised fine-tuning, DEEM can perform various multimodal tasks in a unified manner, such as visual question answering, text-to-image generation, and region-level image captioning.

• **Comprehensive Experiments.** We provide abundant qualitative and quantitative comprehensive experimental results to demonstrate the effectiveness and efficiency of the proposed method.

## 2 METHOD

In this section, we first present our DEEM , starting with an introduction to the overall architecture in Section 2.1, followed by a description of the pipeline in Section 2.2. Finally, we provide details on the training and inference process in Section 2.3.

### 2.1 ARCHITECTURE

In this subsection, we present the multi-modal architecture for processing interleaved image-text data. To excel in both comprehension and creation tasks of text and images, a multi-modal model consists of the following three key components.

**VFM-based Image Encoder** $\mathcal{E}_V$ which encodes each image $x^V \in \mathbb{R}^{H \times W \times 3}$ into an image embedding $e^V \in \mathbb{R}^{N \times C}$, where $C$ is the channel dimension and $N$ is the number of visual tokens in image embedding. **LLM-based Multi-modal Decoder** $\mathcal{D}_{\mathbf{LLM}}$ that extracts context features from the interleaved image-text token sequences. Its input sequence $E \in \mathbb{R}^{K \times C}$ is a concatenation of

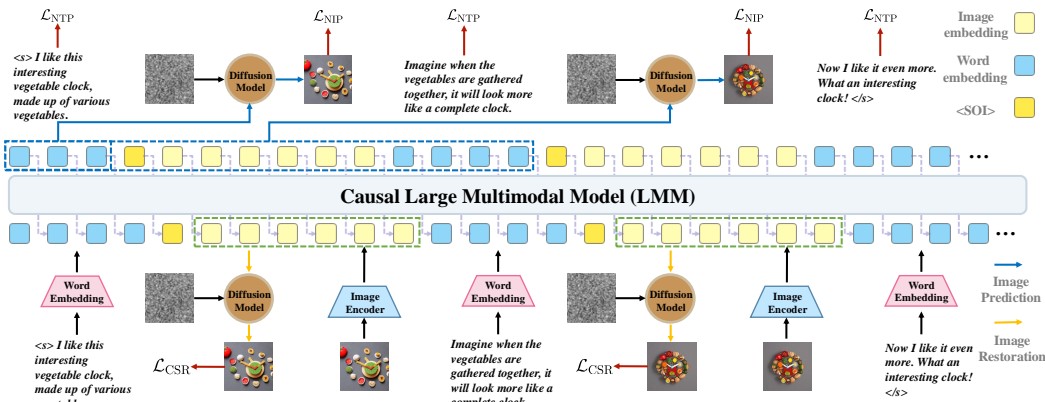

Figure 2: **Overview of our DEEM framework**. Interleaved documents serve as input, decoded to produce outputs. Both text and images are encoded into sequential, discrete token embeddings for the LMM input. Here, we replace the  token embedding in the text with the image embedding before inputting it into the LLM. The text is predicted in an autoregressive manner and the images are synthesized by the DM-based image decoder conditioned on holistic historical semantics captured by LMM. Besides, the image token embeddings are fed into DM-based image decoder for consistent image restoration. The start of image token <SOI> is used to determine the starting position of the image, facilitating the natural autoregressive generation of interleaved text-image layouts. Note that our core architecture is presented without the connectors between modules for simplicity.

embeddings $(e_1, e_2, \dots)$, where $e_n$ is either a word embedding $e_n^L \in \mathbb{R}^{1 \times C}$ or an image embedding $e_n^V \in \mathbb{R}^{N \times C}$. $K$ is the total number of input tokens. **DM-based Image Decoder** $\mathcal{D}_{\textbf{DM}}$ that generates the image conditioned on image-text sequences context feature.

To provide the conditional inputs for $\mathcal{D}_{\text{DM}}$ and reduce the number of visual tokens in image embedding $e^V$, two different Perceiver Resampler (Alayrac et al., 2022) are employed to map the output features from multi-modal decoder $\mathcal{D}_{\text{LLM}}$ and image encoder $\mathcal{E}_V$ to a fixed number of conditional tokens, respectively. Additionally, we utilize an extra mask-aware visual extractor $\mathcal{E}_{\text{M}}$ for extracting region visual information from image embedding $e^V$ via simple mask-aware operation $\mathcal{E}_{\text{M}}(e^V, \mathcal{M}^V)$, where $\mathcal{M}^V$ is the corresponding binary mask of image $x^V$.

## 2.2 PIPELINE

As shown in Fig. 2, given an interleaved image-text sequence $X = \{x_1, x_2, x_3, \dots\}$, where each element $x_n$ is either a text token (denoted as $x_n^L$) or a whole image (denoted as $x_n^V$). Text and images are arranged in the order in which they appear in the original content. To build an end-to-end generative model for interleaved image-text data, a common practice is to first extract embedding for each text token and each image and then feed them into LLMs, $i.e.$, $e_n^L = \mathcal{E}_L(x_n^L)$ and $e_n^V = \mathcal{E}_{\text{M}}(\mathcal{E}_V(x_n^V), \mathcal{M}_n^V)$, where $\mathcal{E}_L$ denotes word embedding in LLM. $\mathcal{E}_V$ is typically an image encoder followed by a Perceiver Resampler (Alayrac et al., 2022) to map each image to a fixed number of visual tokens. As shown in Fig. 3, we introduce a mask-aware visual extractor $\mathcal{E}_{\text{M}}$ for extracting region visual information from image embedding $e_n^V$ via simple mask-aware operation $\mathcal{E}_{\text{M}}(e_n^V, \mathcal{M}_n^V)$, where $\mathcal{M}_n^V$ is the corresponding binary mask of image $x_n^V$ and the default value is 1. Then, the interleaved generative modeling is trained to maximize the log-likelihood:

$$\log p(X) = \sum_n \log p(x_n|e_{<n}) = \sum_{n \in \mathcal{I}_L} \underbrace{\log p(x_n^L|e_{<n})}_{\text{text prediction}} + \sum_{n \in \mathcal{I}_V} \underbrace{\log p(x_n^V|e_{<n})}_{\text{image prediction}}, \quad (1)$$

where $\mathcal{I}_L$ and $\mathcal{I}_V$ represent the index sets for text tokens and images, respectively. That $< n$ in the subscript represents the abbreviation of $\{1, 2, \dots, n-1\}$. The following paragraphs provide explanations of Eq. (1).

**Text Generation with Multi-modal Condition.** $\log p(x_n^L|e_{<n})$ is similar to traditional causal language modeling, except that the condition also includes previous images. Recent works (Alayrac et al., 2022; Li et al., 2023a; Liu et al., 2024a) have demonstrated the effectiveness of using LLMs

for processing additional visual inputs. The loss function for text generation is

$$\mathcal{L}_{\text{NTP}}(x_n^L|e_{<n}) = -\log p(x_n^L|\mathcal{D}_{\text{LLM}}(e_{<n})), \tag{2}$$

where $\mathcal{D}_{\text{LLM}}$ denotes the LLM network.

**Image Generation with Multi-modal Condition.** Maximizing $\log p(x_n^V|e_{<n})$ aligns with the diffusion denoising process, which recently achieved widespread success in image generation. Maximizing the log-likelihood is derived as minimizing the diffusion modeling loss as

$$\mathcal{L}_{\text{NIP}}(x_n^V|e_{<n}) = \mathbb{E}_{\epsilon,t} ||\epsilon - \mathcal{D}_{\text{DM}}(x_{n,t}^V, t, \mathcal{D}_{\text{LLM}}(e_{<n}))||^2, \tag{3}$$

where $\mathcal{D}_{DM}$ is the diffusion model for denoising process. That $x_{n,t}^V$ is the noisy version of the original image at the denoising step $t$, and the denoising network $\mathcal{D}_{DM}$ is trained to predict the noise $\epsilon$.

**Consistency Semantic Regularization.** In addition to the above text and image generation loss functions, we propose a new consistency semantic constraint term. This term reuses the diffusion model to perform generative checks on the image semantic information extracted by the image encoder, ultimately correcting erroneous knowledge in the pre-trained image encoder. This significantly enhances the out-of-distribution generalization and reduces visual hallucinations in the multi-modal model. The new log-likelihood function can be written as

$$\log p^\star(X) = \sum_{n \in \mathcal{I}_L} \underbrace{\log p(x_n^L|e_{<n})}_{\text{text prediction}} + \sum_{n \in \mathcal{I}_V} \underbrace{\log p(x_n^V|e_{<n})}_{\text{image prediction}} + \sum_{n \in \mathcal{I}_V} \underbrace{\log p(x_n^V|e_n)}_{\text{image restoration}}. \tag{4}$$

Similarly, the corresponding log-likelihood function $\log p(x_n^V|e_n)$ can be equivalently written as the following loss function used in training:

$$\mathcal{L}_{\text{CSR}}(x_n^V|e_n) = \mathbb{E}_{\epsilon,t} ||\epsilon - \mathcal{D}_{\text{DM}}(x_{n,t}^V, t, e_n)||^2. \tag{5}$$

Note that the new end-to-end modeling framework brings significant improvements to the generalization performance of the model without altering the original modeling flexibility or introducing additional modules.

## 2.3 TRAINING AND INFERENCE

We employ a three-stage training process, consisting of image-text alignment pre-training, image-text instruction fine-tuning, and mask-text instruction fine-tuning. Image-text alignment pre-training and image-text instruction fine-tuning are designed to validate the effectiveness and efficiency of semantic consistency regularization in enhancing the visual perception capabilities of LMMs. Mask-text instruction fine-tuning is used to verify whether the model trained with semantic consistency regularization negatively impacts the performance of fine-tuning on downstream tasks in the long term. The image-text alignment pre-training objective is defined as the sum of the next-text prediction loss in Eq. (2), next-image prediction loss in Eq. (3) and consistency semantic regularization loss in Eq. (5)

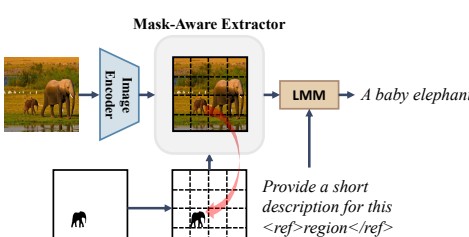

Figure 3: **Pipeline of Mask-Aware Extractor**. The mask-aware extractor can be used to extract region-level visual features based on the mask-aware operation. A simple dot product is applied between the mask and the image embedding before being fed into the LLM.

as $\mathcal{L}_{S_1} = \mathcal{L}_{\text{NTP}} + \lambda \mathcal{L}_{\text{NIP}} + \lambda \mathcal{L}_{\text{CSR}}$, where $\lambda$ is a coefficient used to determine the relative loss weight between the image and text decoding branches. In order to enable the DEEM to perform general multimodal comprehension and creative tasks following human instructions, we use $\mathcal{L}_{S_2} = \mathcal{L}_{\text{NTP}} + \lambda \mathcal{L}_{\text{CSR}}$ to conduct image-text instruction fine-tuning. To further enhance the model's fine-grained region awareness, we conducted region-level mask-text instruction fine-tuning. Since there is no need to perform text-to-image tasks, we removed the next-image prediction loss and the training objective in mask-text instruction fine-tuning can be defined as $\mathcal{L}_{S_3} = \mathcal{L}_{\text{NTP}}$. The whole framework can be optimized end-to-end during the three stages. During inference, the images and texts are generated in an auto-regressive manner. Text tokens are sampled from the distribution predicted by the multi-modal LLM. When the generated token is <SoI>, the diffusion model is called for generating the next image.

## 3 EXPERIMENT

### 3.1 IMPLEMENTATION DETAILS

In this subsection, we first introduce the network of DEEM and then showcase the three-stage training recipes. More details of datasets and hyper-parameters can be found in Table 11.

**Network.** Similar to previous work, We leverage Vicuna7B (Zheng et al., 2024) and Stable Diffusion v2.1 (Rombach et al., 2022) as the large language model, and image decoder, respectively. However, unlike their use of a 427M parameter CLIP-ViT-L as the image encoder, we use a smaller 122M parameter CLIP-ConvNeXt-B(Liu et al., 2022). For the multi-modal LLM, two different Perceiver Resamplers (Alayrac et al., 2022) are used to connect diffusion model with image encoder and large language model respectively.

**Image-Text Alignment Pre-training.** Our model is pre-trained on a mixture of image-text pairs and interleaved image-text sequences, including MMC4-Core (Zhu et al., 2024), LAION-400M (Schuhmann et al., 2021), SBU (Ordonez et al., 2011), and CC-12M (Changpinyo et al., 2021). For LAION-400M (Schuhmann et al., 2021), SBU (Ordonez et al., 2011), and CC-12M (Changpinyo et al., 2021), instead of utilizing the original annotations, we use the version filtered by the pre-trained BLIP-2 model (Li et al., 2023a). For simplicity, we refer to it as BLIP-LCS hereafter. "LCS" abbreviates the LAION, CC, and SBU datasets. The sampling probability of MMC4 is twice that of BLIP-LCS. The images are inserted before or after the corresponding text sentence with equal probability. To optimize training efficiency and data utility, multiple image-text pairs or interleaved image-text sequences are concatenated into extended sequences with the maximum context length.

**Image-Text Instruction Fine-tuning.** To enable DEEM to perform general multimodal comprehension tasks following human instructions, we utilize publicly available datasets for image-text instruction fine-tuning, including LLaVA-665K (Liu et al., 2024a), COCO Caption (Chen et al., 2015), VQAv2 (Goyal et al., 2017), TextCaps (Sidorov et al., 2020), OCR-VQA (Mishra et al., 2019), GQA (Hudson & Manning, 2019), OK-VQA (Marino et al., 2019), TextVQA (Singh et al., 2019), and AOK-VQA (Schwenk et al., 2022).

**Mask-Text Instruction Fine-tuning.** At this stage, we use a simple mask-aware visual extractor to capture pixel-level region features and then align mask-based region features with language embeddings. We collect short text and pixel-level mask pairs from the publicly available object-level datasets (COCO (Chen et al., 2015), RefCOCO (Kazemzadeh et al., 2014), RefCOCO+ (Mao et al., 2016), Ref-COCOg (Mao et al., 2016)), part-level datasets (Pascal Part (Chen et al., 2014), Part Ima-

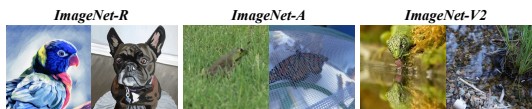

Figure 4: **Examples from ImageNet-R, ImageNet-A, and ImageNet-V2**. These examples share similar backgrounds, rare materials, and unusual textures. They serve as natural adversarial examples and out-of-distribution data, which can be used to test the robustness of models.

genet (He et al., 2022)), and multiple region datasets(VCR (Zellers et al., 2019), Visual Genome (Krishna et al., 2017)). Then we conduct mask-text instruction fine-tuning on the mixture of the above text-mask pairs data, enabling DEEM to complete region-level understanding tasks, such as region-level image captioning.

### 3.2 EXPERIMENTAL RESULTS

In this study, we evaluate our DEEM model by comparing it with current state-of-the-art (SOTA) models on various tasks including visual robustness , hallucination diagnosis, basic visual perception and image-level visual question answering. Please refer to Appendix C for more experimental results about mask-level visual question answering and text-to-image generation. All metrics and data splits are listed in Table 11 in Appendix E.

**Visual Perception Diagnose.** We explore the impact of diffusion feedback on the visual perception capabilities of LMMs from three dimensions: visual robustness, visual hallucinations, and basic visual perception. To rigorously assess visual robustness of our model, we design a benchmark called RobustVQA for robustness evaluation based on online datasets, including ImageNet-A (Hendrycks et al., 2021b), ImageNet-R (Hendrycks et al., 2021a) and ImageNet-V2 (Recht et al., 2019). As shown in Fig. 4, these challenging natural adversarial examples and out-of-distribution samples in

the original ImageNet dataset can be used to evaluate the neural network robustness of our model. Similar to the POPE and MMVP dataset, we first choose the challenging sample from ImageNet-A, ImageNet-R, and ImageNet-V2 dataset and then convert the them into a VQA format that the multimodal model can evaluate simply and accurately. More details about the new benchmark RobustVQA design can be found in Appendix E.1. For a comprehensive visual robustness and hallucination evaluation, we evaluate our model against other open-source state-of-the-art (SOTA) LMMs for text and image generation, including SEED (Ge et al., 2023), SEED-X (Ge et al., 2024), MM-Interleaved (Tian et al., 2024), and DreamLLM (Dong et al., 2023), on the RobustVQA, POPE (Li et al., 2023c) and MMVP (Tong et al., 2024) dataset with accuracy metric. The results, presented in Table 1, demonstrate that our DEEM model not only exhibits competitive performance compared with existing fine-tuned SOTA models on POPE and MMVP after fine-tuning, but also achieves the best results among visual robustness benchmark only after pre-training. Notably, compared to the larger-scale concurrent SOTA model for interleaved text-image modeling, MM-Interleaved (Tian et al., 2024), our model achieves better results with a smaller scale. DEEM outperforms MM-Interleaved 9.4% on RobustVQA, 17.8% on POPE and 9.1% on MMVP. To ensure a fair comparison and prove the effectiveness of our method, we also train an MM-Interleaved model with the same experimental setting as a baseline. Compared to this baseline, Our method achieves an 4% average gain on RobustVQA, 12.8% average gain on POPE and 6.5% average gain on MMVP, respectively. The experimental results demonstrate the effectiveness of our method for better LMMs' visual perception capability.

**Image-Level Visual Question Answering and Captioning.** In order to assess multimodal vision and language capabilities of DEEM , we conduct evaluation against current SOTA LMMs including LLaVA-1.5 (Liu et al., 2023), Qwen-VL (Bai et al., 2023), DreamLLM (Dong et al., 2023) and MM-Interleaved (Tian et al., 2024) across several tasks, including image captioning on COCO (Chen et al., 2015), Image2Paragraph (Krause et al., 2017), visual question answering on VQAv2 (Goyal et al., 2017), OKVQA (Marino et al., 2019), GQA (Hudson & Manning, 2019), VizWiz (Gurari et al., 2018), and VisDial (Das et al., 2017). As demonstrated in Table 2, DEEM exhibits superior or comparable performance relative to SOTA models. In comparison with models for text generation only, our approach consistently achieves competitive performance across various dataset splits. Against models for both image and text generation, DEEM demonstrates enhanced performance in nine dataset splits. Compared to the current state-of-the-art model DreamLLM, DEEM outperforms DreamLLM in six out of the seven shared evaluation dataset splits. It is noteworthy that DEEM is trained with a significantly smaller image encoder CLIP-ConvNeXt-B (Liu et al., 2022), comprising only 122M parameters, in stark contrast to baselines such as DreamLLM (Dong et al., 2023), which utilize larger 427M CLIP-ViT-L (Radford et al., 2021). These results indicate that our method can enhance the model's robustness performance without compromising the multimodal vision and language capabilities of our model.

Table 1: **Zero-shot visual robustness, hallucination and perception evaluation** of RobustVQA-A: RVQA-A, RobustVQA-R: RVQA-R, RobustVQA-V: RVQA-V, POPE-Random: POPE-R (Li et al., 2023c), POPE-Popular: POPE-P (Li et al., 2023c), POPE-Adversarial: POPE-A (Li et al., 2023c) and MMVP (Tong et al., 2024) benchmarks. RobustVQA-A, RobustVQA-R, and RobustVQA-V are robustness benchmarks designed by us in Appendix E.1. "AVG" denotes the overall average accuracy of seven benchmarks. "SFT" denotes the supervised fine-tuning. "*" denotes baseline model without diffusion feedback. The evaluation metrics for each benchmark are listed in Table 12.

| Method | SFT | Architecture | RVQA-A | RVQA-R | RVQA-V | POPE-R | POPE-P | POPE-A | MMVP | AVG |
|---|---|---|---|---|---|---|---|---|---|---|
| *Models for Text-Generation Only* | | | | | | | | | | |
| Shikra (Chen et al., 2023) | ✓ | ViT-L/LLaMA 7B | 33.71 | 38.33 | 37.45 | 86.90 | 83.97 | 83.10 | 22.56 | 55.15 |
| NeXT-Chat (Zhang et al., 2023a) | ✓ | ViT-L/Vicuna 7B | 44.82 | 43.67 | 47.30 | 87.70 | 84.57 | 81.93 | 27.41 | 59.62 |
| *Models for Text and Image Generation* | | | | | | | | | | |
| MM-Interleaved (Tian et al., 2024) | ✗ | ViT-L/Vicuna 13B | 50.76 | 52.71 | 50.60 | 64.73 | 65.33 | 65.20 | 23.82 | 53.31 |
| Emu-I (Sun et al., 2023b) | ✓ | ViT-L/Vicuna 7B | 46.40 | 49.12 | 47.36 | 61.28 | 56.79 | 56.01 | 22.69 | 48.52 |
| SEED (Ge et al., 2023) | ✓ | ViT-G/Vicuna 7B | 52.06 | 59.71 | 57.02 | 69.84 | 56.83 | 59.63 | 25.62 | 54.39 |
| DreamLLM (Dong et al., 2023) | ✓ | ViT-L/Vicuna 7B | 51.43 | 58.96 | 57.60 | 86.36 | 80.07 | 72.63 | 26.37 | 61.84 |
| SEED-X (Ge et al., 2024) | ✓ | ViT-G/Vicuna 13B | 52.36 | 60.27 | 59.49 | 86.41 | 81.43 | 74.56 | 29.16 | 63.39 |
| DEEM * | ✗ | ConvNext-B/Vicuna 7B | 53.24 | 56.06 | 54.72 | 50.55 | 52.00 | 51.93 | 20.30 | 48.40 |
| DEEM | ✗ | ConvNext-B/Vicuna 7B | **56.86** | **68.63** | **63.08** | 69.93 | 70.27 | 68.87 | 28.74 | 60.91 |
| DEEM -VQA | ✓ | ConvNext-B/Vicuna 7B | 55.22 | 64.12 | 62.99 | **87.40** | **82.80** | **78.49** | **32.89** | **65.56** |

Table 2: **Multi-modal comprehension evaluation**. "ED" denotes using extra in-house data. Benchmarks include COCO (Chen et al., 2015); I2Para.: Image2Paragraph (Krause et al., 2017); VQA$^{v2}$: VQAv2 (Goyal et al., 2017); OKVQA (Marino et al., 2019); GQA (Hudson & Manning, 2019); VizWiz (Gurari et al., 2018); VisDial (Das et al., 2017); MMBench: MMB (Yu et al., 2024); MMVet (Yu et al., 2024);. The evaluation metrics for each benchmark are listed in Table 12.

| Model | LLM | VFM | ED | COCO | I2Para. | VQA$^{v2}$ | OKVQA | GQA | VizWiz | VisDial | MMB | MMVet |
|---|---|---|---|---|---|---|---|---|---|---|---|---|
| *Models for Text-Generation Only* | | | | | | | | | | | | |
| IDEFICS-80B (IDEFICS, 2023) | LLaMA-65B | ViT-H | ✗ | 91.8 | – | 60.0 | – | 45.2 | 36.0 | – | 27.9 | – |
| IDEFICS-80B-I (IDEFICS, 2023) | LLaMA-65B | ViT-H | ✗ | 117.2 | – | 37.4 | – | – | 26.0 | – | – | – |
| KOSMOS-1 (Huang et al., 2024) | MetaLM | ViT-L | ✓ | – | – | 46.7 | – | – | – | – | – | – |
| KOSMOS-2 (Peng et al., 2023) | KOSMOS-1 | ViT-L | ✓ | – | – | 45.6 | – | – | – | – | – | – |
| Flamingo-9B (Alayrac et al., 2022) | Chinchilla-7B | ViT-L | ✓ | 79.4 | – | 51.8 | 44.7 | – | 28.8 | 48.0 | 7.9 | 23.3 |
| Flamingo-80B (Alayrac et al., 2022) | Chinchilla-70B | ViT-H | ✓ | 84.3 | – | 56.3 | 50.6 | – | 31.6 | 52.0 | – | – |
| mPLUG-DocOwl (Ye et al., 2023) | LLaMA-7B | ViT-L | ✗ | 52.6 | – | – | – | – | – | – | 60.8 | 35.7 |
| BLIP-2 (Li et al., 2023a) | Vicuna-7B | ViT-L | ✗ | – | – | – | – | 38.6 | 25.3 | – | – | – |
| BLIP-2 (Li et al., 2023a) | Vicuna-13B | ViT-L | ✗ | – | – | 41.0 | – | 41.0 | 19.6 | – | – | – |
| InstructBLIP (Dai et al., 2024) | Vicuna-7B | ViT-L | ✗ | – | – | – | – | 49.2 | 34.5 | – | 68.9 | 33.1 |
| InstructBLIP (Dai et al., 2024) | Vicuna-13B | ViT-L | ✗ | – | – | – | – | 49.5 | 33.4 | – | – | – |
| Shikra (Chen et al., 2023) | Vicuna-13B | ViT-L | ✗ | 117.5 | – | 77.4 | – | – | – | – | – | – |
| LLaVA-1.5 (Liu et al., 2023) | Vicuna-7B | ViT-L | ✗ | – | – | 78.5 | – | 62.0 | 50.0 | – | 53.1 | 32.9 |
| LLaVA-1.5 (Liu et al., 2023) | Vicuna-13B | ViT-L | ✗ | – | – | 80.0 | – | 63.3 | 53.6 | – | 60.6 | 35.6 |
| Qwen-VL (Bai et al., 2023) | Qwen-7B | ViT-G | ✗ | – | – | 78.8 | – | 59.3 | 35.2 | – | 32.9 | 13.0 |
| Qwen-VL-Chat (Bai et al., 2023) | Qwen-7B | ViT-G | ✓ | – | – | 78.2 | – | 57.5 | 38.9 | – | 59.1 | – |
| *Models for both Image and Text Generation* | | | | | | | | | | | | |
| CM3Leon (Yu et al., 2023) | – | – | ✓ | 61.6 | 10.5 | 47.6 | 23.8 | – | 37.6 | 22.6 | – | – |
| Emu (Sun et al., 2023b) | Vicuna-13B | ViT-L | ✓ | 112.4 | – | 52.0 | 38.2 | – | 34.2 | 47.4 | – | – |
| Emu-I (Sun et al., 2023b) | Vicuna-13B | ViT-L | ✓ | **117.7** | – | 40.0 | 34.7 | – | 35.4 | 48.0 | – | – |
| Emu2 (Sun et al., 2023a) | LLaMA-33B | ViT-L | ✓ | – | – | 33.3 | 26.7 | – | 40.4 | – | – | – |
| DreamLLM (Dong et al., 2023) | Vicuna-7B | ViT-L | ✗ | 103.7 | 8.4 | **72.9** | 52.2 | – | 49.3 | – | 58.2 | 36.6 |
| DEEM -VQA | Vicuna-7B | ConvNext-B | ✗ | 115.4 | **22.4** | 68.2 | **53.4** | **55.7** | 50.4 | 42.1 | **60.8** | **37.4** |

## 3.3 ABLATION STUDY

In this study, we conduct ablation studies on several key components of the model, including consistency semantic regularization, training latency, scalability and the impact of different architectures. Benchmarks include RobustVQA-A:RVQA-A; RobustVQA-R: RVQA-R; RobustVQA-V:RVQA-V; POPE-R (Li et al., 2023c); POPE-P (Li et al., 2023c); POPE-A (Li et al., 2023c); MMVP (Tong et al., 2024); OK-VQA (Marino et al., 2019). More additional ablation studies can be found in Appendix D.

**Consistency Semantic Regularization and Training Latency.** To evaluate the effectiveness of the key elements of our design, we conduct the following ablation experiments. We first pre-train a baseline model without using the consistency semantic regularization term under the same training setting for comparison to demonstrate the effectiveness of our architecture. As we can see from Table 3, during the pre-training phase, using our consistency semantic regularization can significantly enhance the model's performance on both hallucination and robustness benchmarks. Moreover, we load the weights of the pre-trained model for image-text instruction fine-tuning experiments. In the second phase of image-text instruction fine-tuning experiments, we demonstrate the effectiveness of our model design. As detailed in Table 3, we observe that after fine-tuning with image-text instruction data, the model's visual hallucination ability improves further, but its visual perception robustness decreases. However, using our consistency semantic regularization can mitigate the robustness degradation while further enhancing the model's visual hallucination ability. To explore the impact of introducing consistency semantic regularization on the training latency in the two stages

Table 3: **Ablation study of $\mathcal{L}_{\text{CSR}}$ and training latency.** Using semantic consistency regularization during both the pre-training and supervised fine-tuning phases can significantly enhance the model's robustness and resistance to hallucinations, while incurring only a marginal additional training cost.

| SFT | $\mathcal{L}_{\text{CSR}}$ | RVQA-A | RVQA-R | RVQA-V | POPE-R | POPE-P | POPE-A | SPEED |
|---|---|---|---|---|---|---|---|---|
| ✗ | ✗ | 53.2 | 56.1 | 54.7 | 50.6 | 52.0 | 51.9 | **8.11** s/step |
| ✗ | ✓ | **57.8** | **69.0** | **64.8** | **69.9** | **70.3** | **68.9** | 9.25 s/step |
| ✓ | ✗ | 51.3 | 56.5 | 57.4 | 85.4 | 78.8 | 76.2 | **2.14** s/step |
| ✓ | ✓ | **53.5** | **57.6** | **58.1** | **86.0** | **79.2** | **77.1** | 2.22 s/step |

of training, we conduct corresponding ablation experiments. We present the result in Table 3. Employing consistency semantic regularization adds only a marginal increase in training latency, yet it significantly enhances the model's robustness capabilities.

Table 4: **Ablation study of model scalability.** Gradually expanding the training data and model size can further enhance the model's capabilities, demonstrating the scalability of the approach.

| Architecture | Training Data | RVQA-A | RVQA-R | RVQA-V | POPE-R | POPE-P | POPE-A | OK-VQA |
|---|---|---|---|---|---|---|---|---|
| ConvNext-B/Vicuna 7B | 32K | 51.86 | 54.31 | 52.73 | 48.44 | 50.10 | 50.06 | 20.74 |
| ConvNext-B/Vicuna 7B | 96K | 52.31 | 57.43 | 54.06 | 54.42 | 57.22 | 56.35 | 22.33 |
| ConvNext-B/Vicuna 7B | 160K | 52.89 | 58.93 | 55.31 | 60.28 | 60.74 | 59.96 | 23.65 |
| ConvNext-L/Vicuna 7B | 160K | 53.23 | 60.47 | 56.88 | 61.12 | 62.87 | 62.09 | 23.87 |
| ConvNext-B/Vicuna 13B | 160K | **53.92** | **61.27** | **57.02** | **62.60** | **64.26** | **63.19** | **31.11** |

**Model Scalability.** Although DEEM demonstrates better performance with smaller data count and model sizes, its scalability has yet to be validated. As is well known, scalability is crucial for model performance. We conduct ablation experiments to assess the scalability concerning data count and model size. As shown in Table 4, gradually increasing the training data enables the model to successfully scale while achieving improved results. Additionally, increasing the sizes of both the VFM and LLM leads to sustained performance enhancements, indicating that DEEM possesses good scalability.

Table 5: **Ablation study of different architectures.** Our method not only significantly enhances the capabilities of LLMs for text and image generation with marginal additional training costs, but it also improves the performance of LLMs for text generation only, validating the generalization ability of the approach.

| Name | $\mathcal{L}_{\text{CSR}}$ | MMVP | RVQA-A | RVQA-R | RVQA-V | POPE-R | POPE-P | POPE-A |
|---|---|---|---|---|---|---|---|---|
| LLaVA | ✗ | 18.6 | 54.8 | 60.0 | 58.7 | 55.5 | 53.3 | 54.6 |
| LLaVA | ✓ | **25.1** | **56.7** | **66.7** | **61.9** | **67.9** | **68.7** | **65.4** |
| DEEM | ✗ | 20.3 | 53.2 | 56.1 | 54.7 | 50.6 | 52.0 | 51.9 |
| DEEM | ✓ | **28.7** | **56.9** | **68.6** | **63.1** | **69.9** | **70.3** | **68.9** |

**Impact of Different Architectures.** By cleverly reusing the diffusion model from LMMs for image and text generation, we can significantly enhance the model's foundational visual perception, visual robustness, and anti-hallucination capabilities with only marginal additional training costs. However, whether DEEM possesses sufficient generalization ability to remain effective for LMMs on text generation only has yet to be explored. To validate our hypothesis, we employ the LLaVA (Liu et al., 2024a) architecture and conducted ablation experiments using semantic consistency regularization loss, with results presented in Table 5. We observe that utilizing diffusion feedback to improve the basic perceptual capabilities of LMMs—thus preventing the model from overly compressing visual information and losing sensitivity to subtle details—is a general method that is architecture-agnostic and exhibits good generalization properties. This suggests that the benefits of our approach could extend beyond the specific configurations tested, potentially enhancing a wide range of LMMs in various applications.

## 4 RELATED WORK

### 4.1 DIFFUSION MODELS FOR REPRESENTATION LEARNING

Diffusion models have made significant progress in various generative tasks (Song et al., 2020; Ho et al., 2020b), such as image generation (Betker et al., 2023), video generation (Ho et al., 2022), and object tracking (Luo et al., 2023). In addition to the aforementioned research, many studies focus on leveraging diffusion models for representation learning. Some works utilize the conditional control of pre-trained diffusion models to flexibly address different downstream tasks, including object classification (Xiang et al., 2023), semantic segmentation (Xu et al., 2023), image caption (Wei et al., 2024), and keypoint matching (Nam et al., 2023). Other studies (Li et al., 2023b; Song et al., 2024) design specialized modules and train diffusion models from scratch to further enhance representation capabilities. Although diffusion models have been widely applied in the generative tasks of

large multimodal models, the use of diffusion models to optimize the visual representations of large multimodal models has yet to be explored. To our knowledge, we are the first to employ diffusion models in a self-supervised paradigm to optimize the visual representations of large multimodal models, significantly enhancing their perceptual abilities and reliability at minimal cost.

## 4.2 Large Multimodal Model

Image-to-text large multimodal models (LMMs) (Luo et al., 2025; Liu et al., 2024c; Zhang et al., 2023b; Wang et al., 2024; Zhou et al., 2024; Liu et al., 2024b) inject visual information into large language models (LLMs) through vision foundation models (VFMs), allowing the language models to perceive visual inputs and thus generate captions or answer questions based on the given multimodal content. Flamingo (Alayrac et al., 2022) tries to extract vision features with a resampler, and transfer them into the text features with a cross-attention mechanism. Instead of using cross-attention layers, BLIP-2 (Li et al., 2023a) directly feed the visual features into the LLMs as soft prompts and significantly reduce the training cost by reducing the visual token number. LLaVA (Liu et al., 2024a) and MiniGPT-4 (Zhu et al., 2023) construct a small-scale instruction tuning dataset to better align the LMM with the expected output format. Although this unidirectional image-to-text paradigm has achieved tremendous success, it still fails to unify multimodal tasks like text-to-image generation and image-to-text visual question answering, significantly limiting the capabilities of multimodal models.

In order to unify multimodal tasks into a unified manner, some works (Yu et al., 2023; Koh et al., 2024; Sun et al., 2023b; Dong et al., 2023; Tian et al., 2024; Ge et al., 2023; 2024; Luo et al., 2024) attempt to generate images and text in the interleaved context concurrently. The release of some public large-scale interleaved image-text datasets (Laurençon et al., 2024; Zhu et al., 2024) has significantly advanced the development of this field. CM3Leon (Yu et al., 2023) converts images into discrete tokens, facilitating token-level auto-regressive modeling as traditional language modeling. Although CM3Leon showcases competitive image generation capabilities, it exhibits notable weaknesses in image understanding. Emu (Sun et al., 2023b) and DreamLLM (Dong et al., 2023) focus on single-stage end-to-end modeling using raw image pixels as input for interleaved image-text generation modeling, but they feed image information at the input of LMMs, which are limited by the problem that fixed number of visual tokens cannot efficiently describe image details. MM-Interleaved (Tian et al., 2024) addresses this limitation by integrating image details into LMMs via multi-scale visual features. However, when faced with out-of-distribution noisy data, the image encoders used by LMMs often produce incorrect visual information, ultimately leading to erroneous predictions. This significantly limits the application of the models in safety-critical scenarios. Building on an advanced interleaved content modeling mechanism, we propose DEEM , which cleverly reuses DMs to correct the outputs of the VFMs without increasing extra parameter count, thereby enhancing the model's generalization capabilities and reducing visual hallucinations in a self-supervised manner. Similar to previous work (Liu et al., 2024a; Dong et al., 2023; Tian et al., 2024), after supervised fine-tuning, it achieves competitive performance on multiple downstream multimodal tasks with the smallest scale.

## 5 Conclusion

Can diffusion models serve as the eyes of large language models for image perception? In this paper, we answer the question by proposing a novel method called DEEM , which leverages a diffusion model as the eyes for LLMs. This approach enhances the robustness of the multimodal model for interleaved image-text modeling and reduces visual hallucinations without introducing extra modules. Through comprehensive exploratory experiments, we demonstrate the effectiveness of the proposed DEEM method. In addition to its advanced robust performance and visual hallucination handling capabilities, we adopt an additional two-stage instruction fine-tuning process to broaden the application scenarios of our DEEM . This enables DEEM to handle a variety of multimodal tasks, including visual question answering, image captioning, and region-level image reasoning. Besides, this work initiates the first step towards visual robustness via generative feedback in a multimodal model. In the future, we will continue to enhance the model's ability to conduct better multimodal comprehension and creation tasks. As an end-to-end framework, we hope it will spur further research in the multimodal robustness field, such as multimodal agents that can handle complex tasks that require safety abilities.

## 6 ACKNOWLEDGMENTS

Min Yang is supported by National Key Research and Development Program of China (2022YFF0902100), National Natural Science Foundation of China (Grant No. 62376262), the Natural Science Foundation of Guangdong Province of China (2024A1515030166). Xiaobo Xia is supported by MoE Key Laboratory of Brain-inspired Intelligent Perception and Cognition, University of Science and Technology of China (Grant No. 2421002).

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

CONTENTS

## A    LIMITATION

Although our method significantly enhances the visual robustness of interleaved image-text modeling multimodal models after image-text alignment pre-training, it, unfortunately, cannot eliminate but only alleviate the robustness knowledge forgetting issue caused by subsequent fine-tuning, as shown in the Table 3. Additionally, our model requires using a diffusion model as another eye to correct and update the erroneous knowledge of the image encoder to improve the overall visual robustness of the multimodal model. However, updating larger image encoders such as CLIP-ViT-L and CLIP-ViT-G(Radford et al., 2021) will increase the training overhead, which may limit the application of our model. We hope that in the future, the diffusion model can completely replace the image encoder to further enhance the effectiveness of our method.

## B    BROADER IMPACTS

The proposed method introduces a novel strategy to enhance the robustness and generalization capabilities of multimodal models by leveraging a diffusion model as an additional eye for large language models. This strategy allows for the correction and updating of potential semantic errors in the image encoder, leading to significant improvements in handling out-of-distribution data and mitigating visual hallucinations. Overall, our contributions provide a significant step forward in the field of multimodal, offering a robust, efficient, and scalable solution for improving the accuracy and reliability of multimodal models. The broader impacts of this work include the potential for more intelligent and adaptive AI systems that can operate effectively in diverse and challenging environments.

Table 6: **Zero-shot region-level image captioning results** on ReferCOCOg.

| Method | Type | METEOR | CIDEr |
|---|---|---|---|
| GRIT (Wu et al., 2022) | Box | 15.2 | 71.6 |
| Kosmos-2 (0-shot) (Peng et al., 2023) | Box | 12.2 | 60.3 |
| Kosmos-2 (2-shot) (Peng et al., 2023) | Box | 13.8 | 62.2 |
| Kosmos-2 (4-shot) (Peng et al., 2023) | Box | 14.1 | 62.3 |
| NeXt-Chat (Zhang et al., 2023a) | Box | 12.0 | 79.0 |
| DEEM -Mask | Mask | 14.1 | 71.0 |

## C    ADDITIONAL EXPERIMENTS RESULTS

**Region-Level Image Captioning.** In addition to holistic image understanding, we also validate the model's ability to take region-level image captioning. As shown in Fig. 3, we use a mask-aware extractor to obtain region-level visual features and address region-level image captioning tasks. We adopt the RefCOCOg (Mao et al., 2016) validation set and compare it with other state-of-the-art (SOTA) models, including GRIT (Wu et al., 2022), Kosmos-2 (Peng et al., 2023), and NeXt-Chat (Zhang et al., 2023a). The CIDEr (Vedantam et al., 2015) and METEOR are applied as the evaluation metrics. As shown in Table 6, our model is capable of achieving competitive performance on CIDEr and METEOR across all of the compared methods, which shows the superiority of our DEEM .

Table 7: **Zero-shot text-to-image generation FID** on MS-COCO and LN-COCO.

| Method | MS-COCO | LN-COCO |
|---|---|---|
| *Text-to-Image Specialists* | | |
| Retrieval Result | 17.97 | 33.59 |
| DALL-E (Ramesh et al., 2021) | ∼28 | - |
| CogView (Ding et al., 2021) | 27.10 | - |
| CogView2 (Ding et al., 2022) | 24.00 | - |
| Stable Diffusion (Rombach et al., 2022) | 12.43 | 34.26 |
| GLIDE (Nichol et al., 2021) | 12.24 | - |
| Make-A-Scene (Gafni et al., 2022) | 11.84 | - |
| DALL-E 2 (Ramesh et al., 2022) | 10.39 | - |
| Muse-3B (Yang et al., 2019) | 7.88 | - |
| Imagen-3.4B (Saharia et al., 2022) | 7.27 | - |
| Parti-20B (Yu et al., 2022) | 7.23 | 15.97 |
| *Models for both Image and Text Generation* | | |
| CM3-13B (Aghajanyan et al., 2022) | 29.56 | - |
| GILL-8B (Koh et al., 2024) | 12.25 | - |
| Emu-13B (Sun et al., 2023b) | 11.66 | - |
| CM3Leon-7B (Yu et al., 2023) | 10.82 | - |
| DreamLLM-7B (Dong et al., 2023) | 8.76 | 22.42 |
| DEEM -7B (Ours) | 8.89 | 24.13 |

**Text-to-Image Generation**. we evaluate text-conditional image generation on MS-COCO (Lin et al., 2014) and LN-COCO (Pont-Tuset et al., 2020). On MSCOCO, we sample 8 images per text condition and use CLIP-ViT-L (Radford et al., 2021) to rerank based on text-image similarity. CLIP

Table 8: Ablation study of input image resolution and coefficient $\lambda$ with 2k training steps and 16 batch size.

| SFT | resolution | $\lambda$ | RVQA-A | RVQA-R | RVQA-V | POPE-R | POPE-P | POPE-A | OK-VQA |
|---|---|---|---|---|---|---|---|---|---|
| ✗ | 256 | 1 | 51.6 | 52.0 | 49.6 | 48.5 | 50.0 | 50.0 | 18.9 |
| ✗ | 256 | 5 | **51.9** | **54.3** | **52.7** | 48.4 | 50.1 | 50.0 | **20.7** |
| ✗ | 256 | 10 | 51.7 | 52.7 | 51.9 | **48.7** | **50.3** | **50.3** | 20.1 |
| ✓ | 256 | 5 | 51.5 | **59.1** | 57.9 | 85.9 | 77.1 | 76.4 | 38.7 |
| ✓ | 448 | 5 | **52.5** | 57.6 | **58.1** | **86.0** | **79.2** | **77.1** | **41.0** |

reranking is not used for LN-COCO. FID (Heusel et al., 2017) is used to evaluate both datasets. As shown in Table 7, our model shows competitive text-to-image generation compared to existing image and text generation models. See qualitative results on text-to-image synthesis in Fig. 10 in Appendix F.

# D    ADDITIONAL ABLATION STUDY

we provide more ablation studies for DEEM in this section, all of which share the same settings. All the code, models, and data tools will be released soon.

## D.1    ABLATION STUDY OF INPUT IMAGE RESOLUTION

In addition to the aforementioned exploration, we also scale up the input image resolution for performance gain. The performance gain becomes larger when further increasing the input image resolution from 256 to 448 in image-text instruction fine-tuning, as shown in Table 8. Such results indicate our method could better exploit the additional information gained from high resolution. Moreover, we conduct an ablation study on coefficient $\lambda$ in loss function. As shown in Table 8, setting $\lambda = 5$ achieves a better balance between robustness and hallucination empirically.

## D.2    ABLATION STUDY OF TRAINING RECIPES

We also conduct an ablation study to control the trainability of different training modules. As shown in Table 10, we found that freezing the DM (Diffusion Model) while not freezing the VFM (Visual Foundation Model) during training yields the best robustness and hallucination results.

# E    ADDITIONAL IMPLEMENTATION DETAILS

## E.1    DATASET CONSTRUCTION

As shown in Fig. 5, we first convert the original ImageNet-A (Hendrycks et al., 2021b), ImageNet-R (Hendrycks et al., 2021a), and ImageNet-V2 (Recht et al., 2019) data into a VQA format that

Table 9: **Comparison of different VQA formats.** Questions in the yes or no format can well evaluate the performance of the models on the RobustVQA benchmark, while questions in the multiple-choice format are very random, and MM-interleaved tend to output the first option. Therefore, we adopt yes or no format in our experimental settings. More details about the new benchmark RobustVQA design can be found in Appendix E.1.

| Format | Prompt | RobustVQA-A | RobustVQA-R | RobustVQA-V |
|---|---|---|---|---|
| multiple-choice | "What is the main object in this image?" "Chose from the list: [false category label,gt category label]." | 44.88 | 58.88 | 46.86 |
| multiple-choice | "What is the main object in this image?" "Chose from the list: [gt category label,false category label]." | **84.60** | **90.16** | **82.92** |
| yes or no | "Is [gt category label] the main object in this image?" "Please answer yes or no." "Is [false category label] the main object in this image?" "Please answer yes or no." | 50.76 | 52.71 | 50.60 |

the multimodal model can evaluate. Specifically, we use the CLIP-ViT-L model for hard example mining, predicting the incorrect category label with the highest confidence score apart from the ground truth category label. We then use a pre-defined prompt as: "`Is [category label] the main object in this image? Please answer yes or no.`" to simultaneously construct a pair of positive and negative example samples, allowing the model to answer "`yes`" or "`no`". By using this design, we can evaluate the robustness of multimodal models in an unbiased manner with the new benchmark called RobustVQA, facilitating both assessment and comparison. It is worth noting that, as shown in Table 9, we find that the yes or no format is more stable than the multiple-choice format and can better evaluate the robustness of multi-modal models.

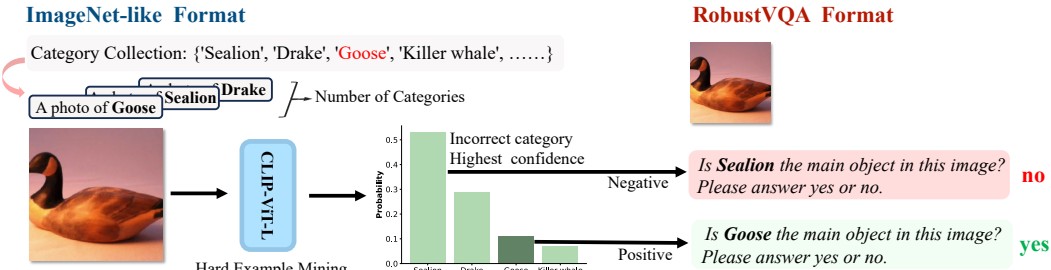

Figure 5: **Robustness dataset construction process.** We use the CLIP-ViT-L model for hard example mining and then transform them into question-answer pairs via a pre-defined template.

### E.2 IMAGE-TEXT ALIGNMENT PRE-TRAINING

We use MMC4-Core (Zhu et al., 2024), LAION-400M (Schuhmann et al., 2021), SBU (Ordonez et al., 2011), and CC-12M (Changpinyo et al., 2021) as the pre-training dataset. For LAION-400M (Schuhmann et al., 2021), SBU (Ordonez et al., 2011), and CC-12M (Changpinyo et al., 2021), instead of

Table 10: Ablation study of training recipe in image-text alignment pre-training with 10k training steps and 128 batch size.

| VFM | DM | RVQA-A | RVQA-R | RVQA-V | POPE-R | POPE-P | POPE-A |
|---|---|---|---|---|---|---|---|
| freeze | unfreeze | 53.2 | 56.1 | 54.7 | 50.6 | 52.0 | 51.9 |
| unfreeze | freeze | **56.8** | **68.6** | **63.1** | **69.9** | **70.3** | **68.9** |
| unfreeze | unfreeze | 50.3 | 52.5 | 53.1 | 54.8 | 56.3 | 56.1 |

utilizing the original annotations, we use the version filtered by the pre-trained BLIP-2 model (Li et al., 2023a). For simplicity, we refer to it as BLIP-LCS hereafter. "LCS" abbreviates the LAION, CC, and SBU datasets. Text prompts with lengths shorter than 10 are also filtered out. Due to network constraints, we only collect approximately 6M of MMC4-Core and 20M of BLIP-LCS data. The sampling probability of MMC4 is twice that of BLIP-LCS. The images are inserted before or after the corresponding text sentence with equal probability. Specifically, images with a CLIP similarity score below 0.24 will be discarded, and only 6 images at most will be kept for each document in MMC4-Core. We also exclude 100% of all documents that do not contain any images, and 50% of documents that contain only 1 image. For image-text-pair BLIP-LCS datasets, we randomly sample multiple image-text pairs from the same dataset and concatenate them to the maximum context length (*i.e.*, 2048) during pre-training. For interleaved image and text MMC4-Core (Zhu et al., 2024) datasets, we also split and concatenate the documents to form the training samples. Such a concatenation strategy can utilize the full context window of Large Language Models and thus achieve high data efficiency. Besides that, for image generation, we ignore the training loss of images which are the first element in the sequence. The text condition of the rest images is dropped with a 10% probability to improve classifier-free guidance sampling. The detailed hyper-parameters of image-text alignment pre-training are listed in Table 11.

### E.3 IMAGE-TEXT INSTRUCTION FINE-TUNING

We utilize public available datasets for supervised fine-tuning, including LLaVA-665K(Liu et al., 2024a), COCO Caption (Chen et al., 2015), VQAv2 (Goyal et al., 2017),TextCaps (Sidorov et al., 2020), OCR-VQA (Mishra et al., 2019), GQA (Hudson & Manning, 2019), OK-VQA (Marino et al., 2019), TextVQA (Singh et al., 2019), and AOK-VQA (Schwenk et al., 2022). We use the following prompt template ``Based on the image, please answer the question. {image} {question}. The answer is: {answer} " to convert the data into a mix-

ture of instruction following forms, resulting in approximately 800K instruction data for the second-stage image-text instruction fine-tuning. The detailed hyper-parameters of image-text instruction fine-tuning are listed in Table 11.

### E.4 MASK-TEXT INSTRUCTION FINE-TUNING

We collect short text and pixel-level mask pairs from the publicly available object-level datasets (COCO, RefCOCO, RefCOCO+) and part-level datasets (Pascal Part, Part Imagenet), then transform them into instruction following data. Moreover, Visual Genome (VG) and Visual Commonsense Reasoning (VCR) datasets are employed to add more multiple region understanding data, resulting in approximately 200K instruction data for the third-stage mask-text instruction fine-tuning. See more hyper-parameters details in Table 11.

### E.5 EVALUATION

As shown in Fig. 6, DEEM achieves the best results on both hallucination and robustness benchmarks even at the smallest scale, demonstrating the efficiency and effectiveness of our approach. In addition to visual robustness and hallucination, we also use various benchmarks and datasets, such as image caption, visual question answering, text-to-image generation and so on, to assess the image-text comprehension capabilities. All these evaluation tasks and metrics are listed in Table 12. The prompt templates for each task are listed in Fig. 8.

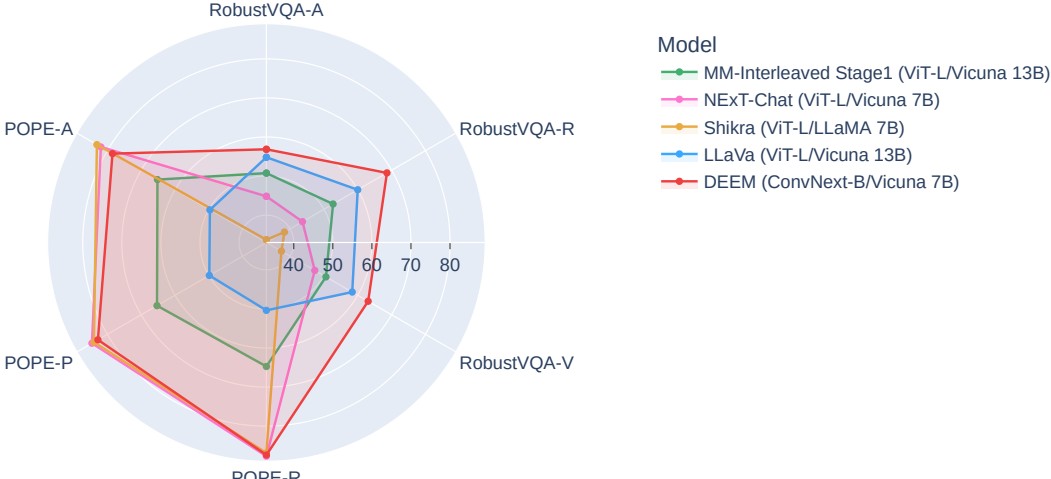

Figure 6: **Performance on visual robustness and hallucination benchmark.** DEEM achieves the best results on robustness benchmark and competitive performance on hallucination even at the smallest scale, demonstrating the efficiency and effectiveness of our approach.

## F ADDITIONAL VISUALIZATION EXAMPLES

### F.1 SEMANTIC IMAGE SYNTHESIS

**Dynamic Semantic Bias Erasure. We demonstrate the dynamic semantic bias elimination process through three iterations on the same sample, providing an illustration of the original image alongside its version reconstructed in real-time according to semantic conditions, as shown in Fig. 9. Our method, DEEM , gradually mitigates potential erroneous semantics within the visual encoder through multiple iterations, ultimately enhancing the perceptual capabilities of MLLMs.**

**Consistency Semantic Image Synthesis** We visualize some consistency semantic image synthesis and display both the original images and their reconstructed versions in Fig. 11. DEEM accurately recovers the features of the original images without causing distortion.

---

**Visual input example, Goose:**

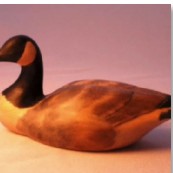

| | |
|---|---|
| User | Based on the image, please answer the question. What is the main object in this image? Chose from the list: ['Sealion', 'Drake', 'Goose', 'Killer whale']. |
| | Let's think step by step. |
| DEEM (Ours) | A carved goose sitting on a flat surface. So the answer is goose ✓ |
| LLaVA | The main object in this image is a drake. ✗ |
| NExT-Chat | A sculpture of a duck sitting on a table. So the answer is duck. ✗ |
| MM-Interleaved | Sealion ✗ |

Figure 7: **Case Comparison.** Compared to other SOTA models, including LLaVA, NeXt-Chat, and MM-Interleaved, when encountering out-of-distribution data, their models are affected by incorrect semantics from the image encoder and cannot output the correct answer. However, DEEM can output the correct answer via generative feedback.

Table 11: **Training recipes** for DEEM . The three training stages are introduced in Section 2.3. `Stage I`: Image-Text Alignment Pre-training, `Stage II`: Image-Text Instruction Fine-tuning, `Stage III`: Mask-Text Instruction Fine-tuning.

| | Stage I | Stage II | Stage III |
|---|---|---|---|
| Phase | Image-Text Alignment | Image-Text SFT | Mask-Text SFT |
| *Training Hyper-Parameters* | | | |
| Input image resolution | 256×256 | 448×448 | 448×448 |
| Output image resolution | 512×512 | 512×512 | 512×512 |
| VFM | CLIP-ConvNext-B | CLIP-ConvNext-B | CLIP-ConvNext-B |
| LLM | Vicuna-7B v1.5 | Vicuna-7B v1.5 | Vicuna-7B v1.5 |
| DM | Stable Diffusion v2.1 | Stable Diffusion v2.1 | Stable Diffusion v2.1 |
| $\lambda$ | 5 | 5 | 5 |
| Learning Rate | 2e-5 (image encoder&decoder) 1e-4 (others) | 1e-6 (language model) 1e-5 (others) | 1e-6 (language model) 1e-5 (others) |
| Optimizer | AdamW | AdamW | AdamW |
| Optimizer hyper-parameters | $\beta_1, \beta_2, \epsilon = 0.9, 0.995, 1e-6$ | $\beta_1, \beta_2, \epsilon = 0.9, 0.999, 1e-8$ | $\beta_1, \beta_2, \epsilon = 0.9, 0.999, 1e-8$ |
| Weight Decay | 0.05 | 0.05 | 0.05 |
| Training iterations | 10k | 10k | 10k |
| Warmup steps | 1k | 500 | 500 |
| Learning Rate Scheduler | Cosine | Cosine | Cosine |
| Batch Size Per GPU | 4 | 16 | 2 |
| Maximum Token Length | 2048 | 2048 | 2048 |
| Augmentation | CenterCrop | - | - |
| Unfreeze LLM | ✗ | ✓ | ✓ |
| Unfreeze DM | ✗ | ✗ | ✗ |
| Unfreeze VFM | ✓ | ✗ | ✗ |
| *Training Data* | | | |
| Dataset | ① MMC4 ② BLIP-LCS | ① LLaVA-Mix-665K ② VQA-Mixture ③ COCO Caption | ① COCO/ReferCOCO/ReferCOCO+ ② Pascal-Part/Part-ImageNet ④ VG/VRC |
| Data Size | ∼26M | ∼800K | ∼200K |
| Data Type | Interleave/Pair | Instruction | Instruction |
| *Training Cost* | | | |
| GPU Device | 32×NVIDIA A100 | 32×NVIDIA A100 | 32×NVIDIA A100 |
| Training Time | ∼30h | ∼6h | ∼3h |

Table 12: **Overall descriptions of the evaluation benchmarks** for evaluating capabilities, including image-level captioning, image-level visual question answering, text-to-image generation, region-level image captioning, visual robustness, comprehension, perception and hallucination.

| | Dataset | Task description | Eval Split | Metric |
|---|---|---|---|---|
| CAP. | COCO (Chen et al., 2015) | Scene description | test | CIDEr(↑) (Vedantam et al., 2015) |
| | Image2Paragraph (Krause et al., 2017) | Scene description | test | CIDEr(↑) (Vedantam et al., 2015) |
| VQA. | VQAv2 (Goyal et al., 2017) | Scene understanding QA | test-dev | VQA Acc(↑) (Antol et al., 2015) |
| | OKVQA (Marino et al., 2019) | External knowledge QA | val | VQA Acc(↑) (Antol et al., 2015) |
| | GQA (Hudson & Manning, 2019) | Scene understanding QA | test-dev | VQA Acc(↑) (Antol et al., 2015) |
| | VizWiz (Gurari et al., 2018) | Scene understanding QA | test-dev | VQA Acc(↑) (Antol et al., 2015) |
| | VisDial (Das et al., 2017) | Image dialogue | val | NDCG(↑) |
| SYN. | MS-COCO (Lin et al., 2014) | Text-Conditional Image Synthesis | val-30K | FID(↓) (Heusel et al., 2017) |
| | LN-COCO (Pont-Tuset et al., 2020) | Text-Conditional Image Synthesis | val | FID(↓) (Heusel et al., 2017) |
| REF. | RefCOCO (Kazemzadeh et al., 2014) | Region-level scene description | val | CIDEr(↑) (Vedantam et al., 2015) |
| | RefCOCO+ (Mao et al., 2016) | Region-level scene description | val | CIDEr(↑) (Vedantam et al., 2015) |
| | RefCOCOg (Mao et al., 2016) | Region-level scene description | val | CIDEr(↑) (Vedantam et al., 2015) |
| OOD. | RobustVQA-V | Out-of-Distribution Robustness | val | Acc(↑) |
| | RobustVQA-R | Out-of-Distribution Robustness | val | Acc(↑) |
| | RobustVQA-A | Out-of-Distribution Robustness | val | Acc(↑) |
| Hall. | POPE-R (Li et al., 2023c) | Visual Hallucination | val | Acc(↑) |
| | POPE-P (Li et al., 2023c) | Visual Hallucination | val | Acc(↑) |
| | POPE-A (Li et al., 2023c) | Visual Hallucination | val | Acc(↑) |
| CPH. | MMBench (Yu et al., 2024) | Visual Comprehension | val | Acc(↑) |
| | MMVet (Yu et al., 2024) | Visual Comprehension | val | Acc(↑) |
| PCP. | MMVP (Tong et al., 2024) | Visual Perception | val | Acc(↑) |

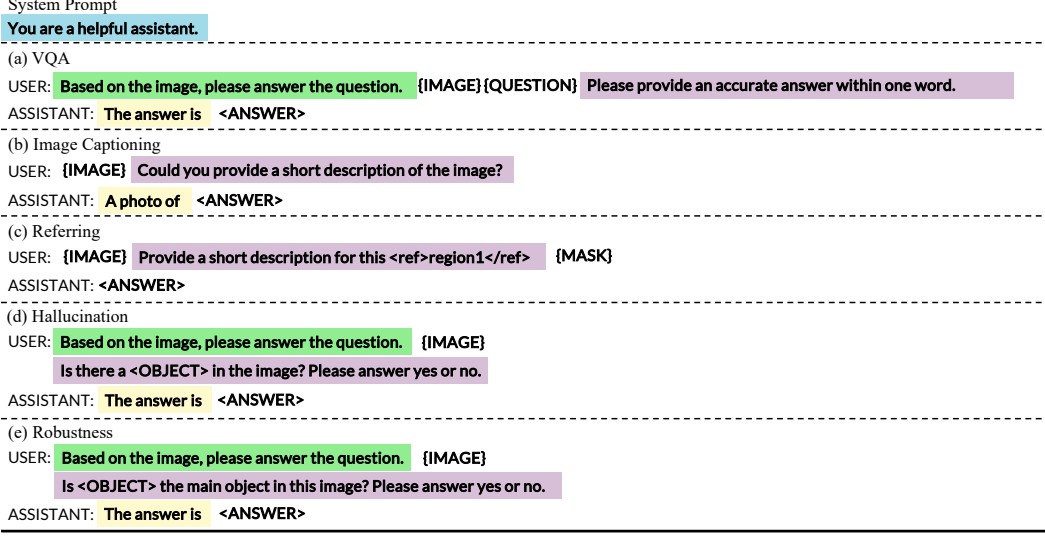

Figure 8: **Prompt template used for evaluation.** (a) VQA includes VQAv2, VizWiz, OKVQA, GQA, VisDial, and MMVP. (b) Image Captioning includes COCO, Image2Paragraph. (c) Region-level Image Captioning includes RefCOCOg. (d) Visual hallucination includes POPE. (e) Visual Robustness includes RobustVQA-A, RobustVQA-R, and RobustVQA-V. < IMAGE >denotes the input image representation, < MASK > denotes the mask-level image representation, < QUESTION >denotes each specific question, < ANSWER > is the generated answer, and < OBJECT > is the specific object name in a question of POPE and RobustVQA.

## F.2 TEXT CONDITION IMAGE SYNTHESIS

In Fig. 10, we present some text-to-image synthesis examples from DEEM , demonstrating its capability to generate corresponding images based on given prompts.

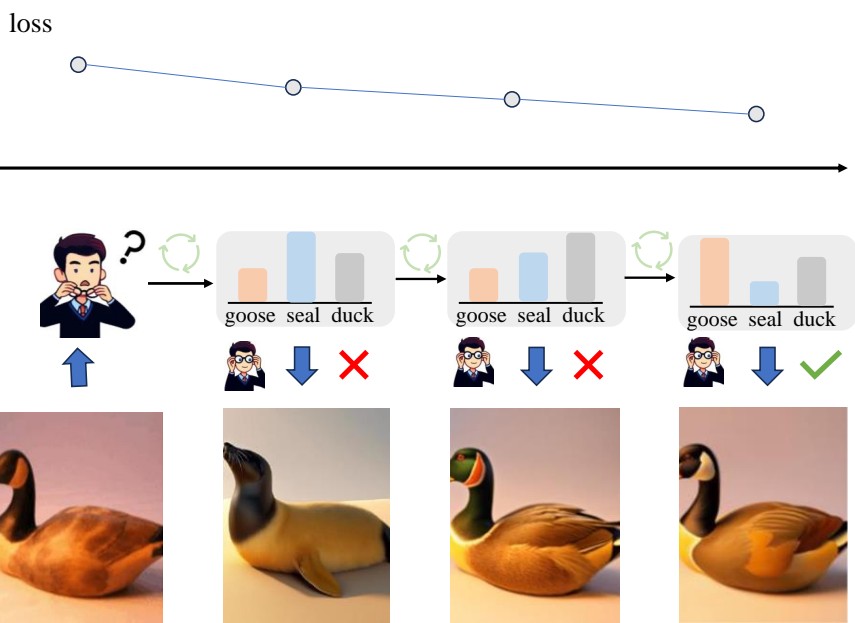

Figure 9: **Dynamic semantic bias elimination process** through three iterations on the same sample, diffusion process is conducted by adding 65% noise to the original image as the initial condition.

### F.3 ROBUSTNESS COMPARISON

In Fig. 7, we present a comparative analysis of visual robustness results between our model, DEEM , and other state-of-the-art models: LLaVA (Liu et al., 2024a), NeXt-Chat (Zhang et al., 2023a), and MM-Interleaved (Tian et al., 2024). When encountering natural adversarial samples or out-of-distribution samples, the image encoder in their models will output incorrect semantic information, leading to incorrect category answers. In contrast, our method uses a diffusion model as the eyes of the large language model to inspect and correct the output features of the image encoder. This process eliminates incorrect semantic outputs from the image encoder, ultimately allowing the large language model to produce the correct category answer. This simple yet effective approach significantly enhances the model's robustness and generalization capabilities.

### F.4 IMAGE-TEXT MULTIMODAL DIALOGUE

In Fig. 12, we show the image-text dialogue case examples of DEEM . Our model can input any interleaved layout of text-image data and simultaneously understand and generate text-image outputs in any interleaved layout, representing the future of next-generation multimodal dialogue.

### F.5 MASK-TEXT MULTIMODAL DIALOGUE

In addition to image-level input, DEEM also supports mask-text input to perform fine-grained region-level reasoning tasks. As shown in the Fig. 13, DEEM can accurately extract region semantics of the image based on the input mask and complete the corresponding instruction tasks.

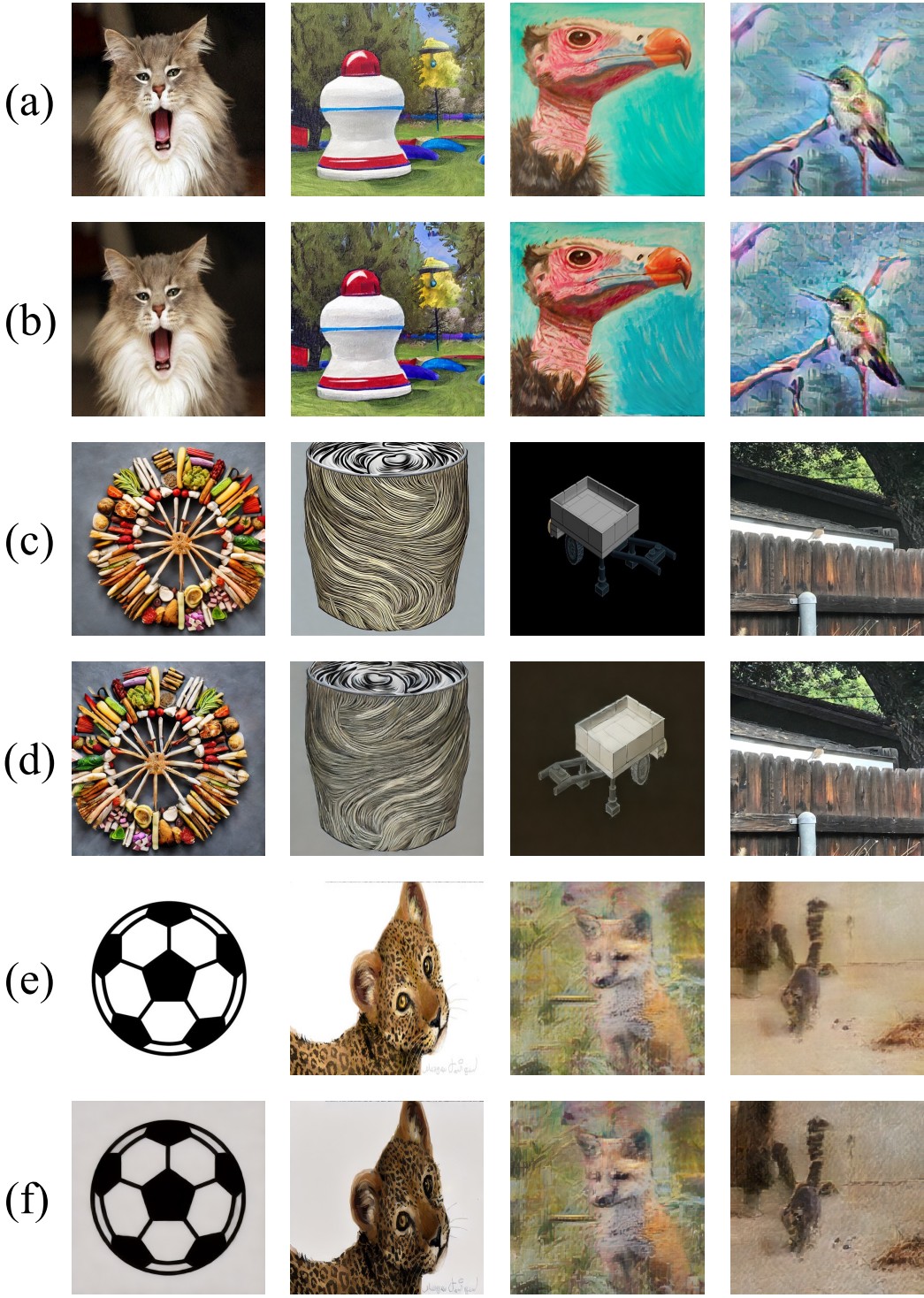

Figure 10: **image-to-image generation examples with the outputs of image encoder.** (a,c,e) are original images and (b,d,f) are synthesis images based on the image embeddings of original images.

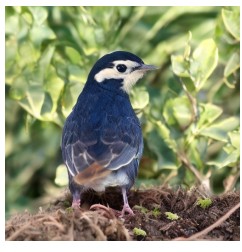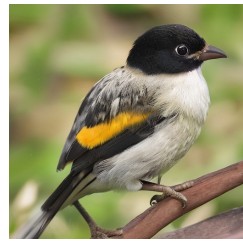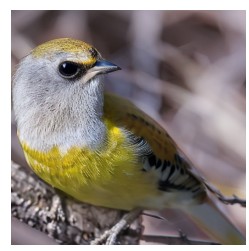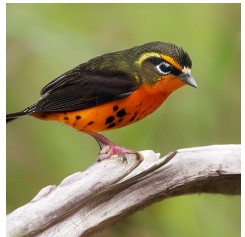

*Small bird with a pale yellow underside light brown crown and back gray tail and wing tips tip of tail feather bright yellow black eyes and black strip over eyes*

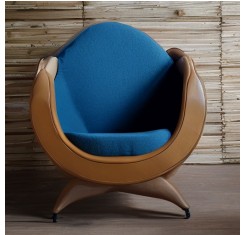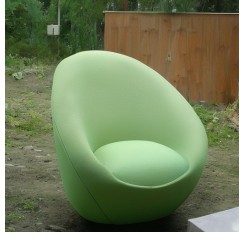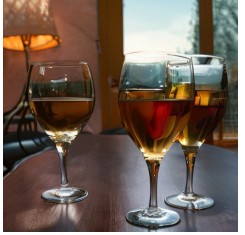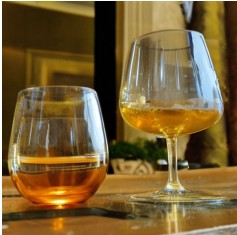

*An armchair in the shape of an avocad*  *A couple of glasses are sitting on a tabl*

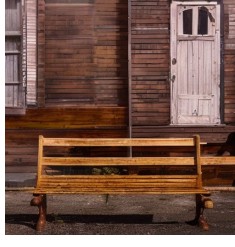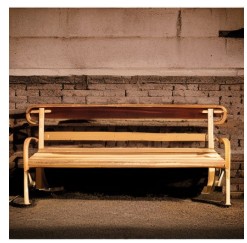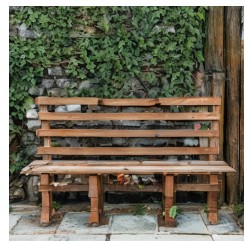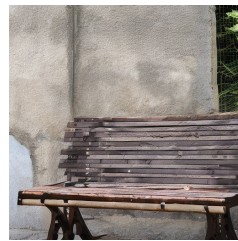

*A long wooden bench in front of a brick wall*

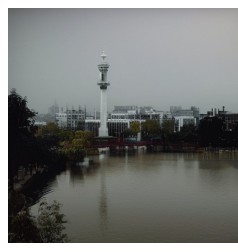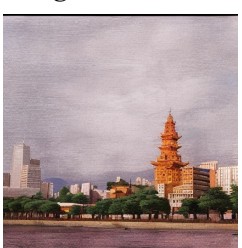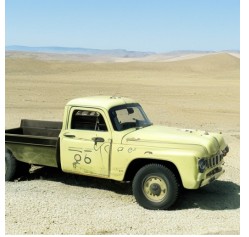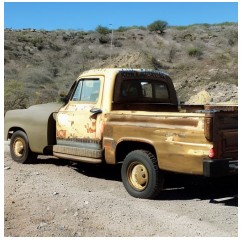

*Downtown Beijing at sunrise. detailed ink wash*  *A beat-up truck in the desert*

Figure 11: **Text-to-image generation examples with prompts.** DEEM can generate vivid images based on input text conditions.

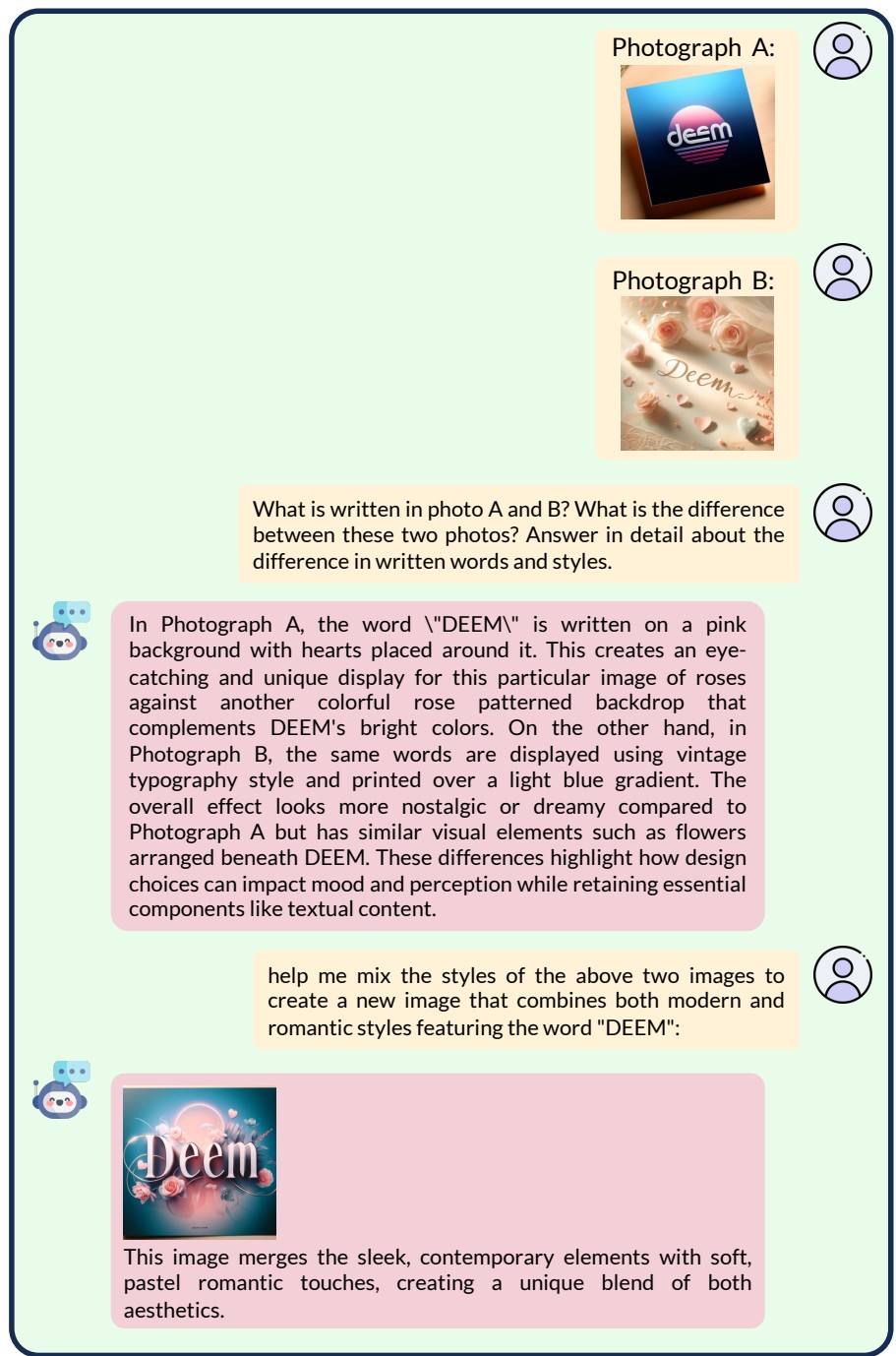

Figure 12: **Examples of image-text multimodal dialogue** between human and DEEM . Text and image can be used as inputs or outputs, and multi-round dialogue is shown.

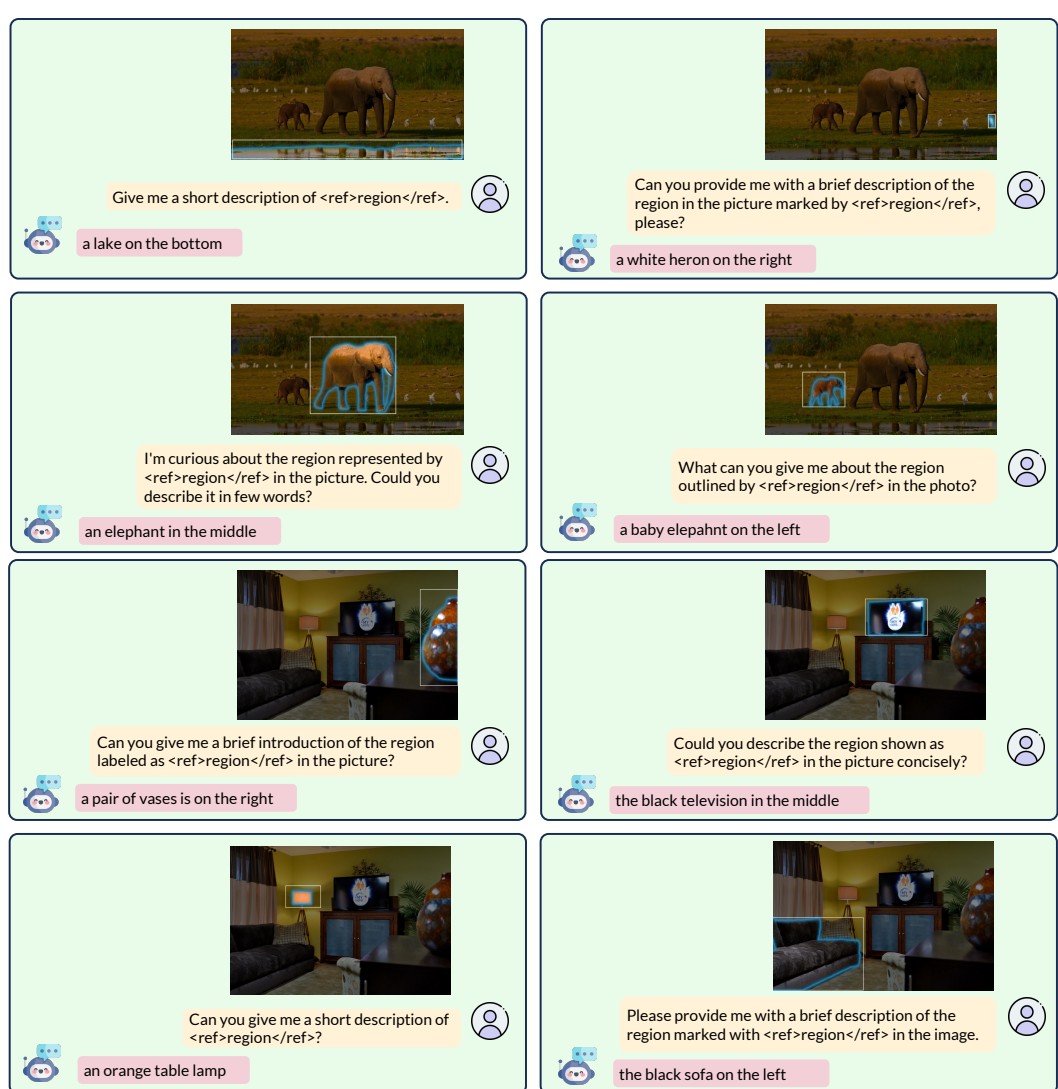

Figure 13: **Examples of mask-text multimodal dialogue** between human and DEEM . Text and mask can be used as inputs and DEEM outputs the corresponding answer, and multi-round dialogue is shown.

