# OpenReview forum: "DEEM: Diffusion models serve as the eyes of large language models for image perception"
_ICLR.cc/2025/Conference — ICLR 2025 Spotlight_

### Official Review · Reviewer_qAMc · 2024-10-27

**Soundness:** 2
**Presentation:** 2
**Contribution:** 2
**Rating:** 6
**Confidence:** 3

**Summary:**

This paper introduces DEEM, a method that leverages the generative feedback of diffusion models to align the semantic distributions of image encoders, addressing issues faced by large multimodal models (LMMs) when encountering out-of-distribution data such as orientation, quantity, color, and structure. DEEM improves the resilience of LMMs against such data and reduces visual hallucinations without the need for additional training modules and with fewer training parameters. It demonstrates significant improvements in visual perception performance across several benchmarks (RobustVQA, POPE, and MMVP), exhibiting enhanced robustness and superior capability to mitigate model hallucinations while using fewer trainable parameters and less pre-training data, and being smaller in base model size. Extensive experiments show that DEEM enhances the performance of LMMs across various downstream tasks, including visual question answering, image captioning, and text-conditioned image synthesis.

**Strengths:**

**Strengths**

1. **Originality**:
   - The paper presents a novel approach, DEEM, which integrates diffusion models to enhance the visual perception capabilities of large language models (LLMs) for multimodal tasks.

2. **Quality**:
   - The paper provides comprehensive empirical evidence to support the effectiveness of the proposed method. The authors perform extensive evaluations on newly constructed and well-known benchmarks (e.g., RobustVQA, POPE, and MMVP), demonstrating the method's robustness and performance improvements.

3. **Clarity**:
   - The paper is well-structured and clearly written, providing a detailed description of the DEEM architecture, training process, and experimental evaluation.

4. **Significance**:
   - DEEM significantly enhances the robustness and visual perception capabilities of LMMs, addressing a critical challenge in the field of multimodal machine learning.

**Weaknesses:**

**Weaknesses**

1. **Scalability Concerns**:
   - While DEEM demonstrates improved performance using fewer training parameters and a smaller base model size, the requirement to employ diffusion models as an additional component could potentially increase computational overhead.
   - As the model size continues to grow, will the computational burden brought by the diffusion models outweigh the performance gains? After all, as the model size increases, its capabilities significantly improve, and computational costs become the primary bottleneck. The introduction of diffusion might bring negative returns.

2. **Robustness Limitation**:
   - Although DEEM significantly enhances visual robustness, the paper acknowledges that the method cannot completely eliminate but only alleviate robustness knowledge forgetting during subsequent fine-tuning stages. This indicates a potential long-term limitation in maintaining the enhanced robustness through the entire lifecycle of model updates and task-specific fine-tuning.

3. **Limited Analysis of Failure Cases**:
   - The paper largely focuses on positive results and improvements brought by DEEM. However, it lacks an in-depth analysis of cases where the method might fail or underperform compared to existing approaches. In particular, in the example shown in figure 9 of the supplementary materials, many generated images exhibit significant errors in color, spatial arrangement, and orientation. These generation flaws somewhat undermine the paper's claim that diffusion models can enhance visual understanding, as diffusion models also cannot perfectly restore visual features.

**Questions:**

**Questions for the Authors**

1. **Scalability and Computational Efficiency**:
   - How does the use of diffusion models as an additional component impact the overall computational efficiency and memory usage, especially for larger versions of image encoders?
   - Have you explored any strategies for optimizing computational resources? If so, could you provide details on these optimization techniques and their effectiveness?

2. **Robustness Knowledge Forgetting**:
   - You mentioned that DEEM cannot completely eliminate robustness knowledge forgetting. Do you have insights or preliminary results on potential methods for further mitigating this issue?

3. **Failure Case Analysis**:
   - Could you provide more details or examples of cases where DEEM underperforms compared to other state-of-the-art methods?
   - What common patterns or factors have you identified in these failure cases, and what potential solutions might address these issues?

4. **Impact of High-Resolution Inputs**:
   - How does the increase in input image resolution from 256 to 448 pixels impact the performance and training dynamics? Are there diminishing returns at higher resolutions?
   - Have you evaluated the impact of image resolution on the computational requirements and time to convergence?

5. **Diffusion Model's Role**:
   - Could you elaborate on how the diffusion model corrects and aligns the semantic outputs of the image encoder during training? Are there specific examples or visualizations that illustrate this process?
   - How critical is the specific architecture or configuration of the diffusion model in achieving these results?

In summary, while the paper proposes combining diffusion models with LLMs to enhance the model's perceptual capabilities, this approach is not entirely novel, as seen in works like Transfusion (https://www.arxiv.org/abs/2408.11039). Additionally, if the sole purpose is to enhance the model's visual perception, introducing such a complex diffusion module may seem excessive. If a diffusion module is introduced, the authors should also focus on evaluating image generation capabilities. Alternatively, the authors should attempt to use simpler methods as baselines to highlight the necessity of introducing the diffusion module. The ablation studies in the paper do not seem to cover this aspect.

---

> ### Author Response · Authors · 2024-11-19
>
> Thanks for the valuable and encouraging comments! Our point-by-point responses to the reviewer's mentioned concerns are provided as follows.
>
> > **W 1&Q1**: Scalability Concerns.
>
> **Response:**
>
> Thank you for your insightful concern. The size of the diffusion model is fixed, and during the scaling of visual encoding and language models, the additional overhead associated with the diffusion model remains constant as its parameters are frozen and not expanded. Therefore, this additional overhead does not increase with the scaling of the visual and language models. As shown in Table 3 of the paper, for DEEM, the additional training overhead introduced by the diffusion model is only 13.9% during the pre-training phase and 3.7% during the fine-tuning phase for each iteration, which is solely related to the average number of images per iteration. Furthermore, the application of inference acceleration methods [1] can significantly reduce the additional overhead by 10%–50%. Consequently, the computational costs will not become the primary bottleneck. Additionally, we can transition from online learning to offline learning, first eliminating erroneous knowledge in the visual encoder and subsequently combining it with the large language model to ultimately produce LLMs with enhanced visual perception capabilities.
>
> [1] Haozhe Liu, et al. "Faster Diffusion via Temporal Attention Decomposition" NeurIPS2024
>
> > **W 2&Q2**:Robustness Limitation
>
> **Response:**
>
> Thank you for your detailed review. We speculate that the phenomenon of Robustness Knowledge Forgetting arises from conflicts in multitask training. Specifically, tasks focused on foundational visual robustness tend to emphasize more global visual information, while Visual Question Answering (VQA) tasks concentrate on more localized visual information. During the training process, a single visual encoder may struggle to effectively handle tasks requiring different levels of visual granularity, leading to the emergence of Robustness Knowledge Forgetting. A viable solution to address the conflicts and resultant knowledge forgetting from joint training of these two tasks is to employ two lightweight visual encoders, as proposed by Janus [1]. One encoder would be dedicated to processing tasks requiring global visual information, while the other would focus on tasks necessitating fine-grained visual details.
>
> [2] Wu, Chengyue, et al. "Janus: Decoupling visual encoding for unified multimodal understanding and generation" NeurIPS2024
>
> > **W 3&Q3**:Limited Analysis of Failure Cases
>
> **Response:**
>
> Thanks for your professional and careful review. In Figure 10 of the supplementary materials, we present our case study, in which we randomly selected 12 images of various types from the test set. These images underwent semantic extraction via a visual encoder and semantic conditional reconstruction using a diffusion model, encompassing both correct and incorrect cases. As illustrated, despite rigorous training on incorrect semantic erasure, certain failure cases still exhibit spatial perception and lighting errors. Our experimental results indicate that more significant semantic biases exist within the native visual encoder. The visual features encoded by the native visual encoder are particularly prone to spatial and color semantic discrepancies. Consequently, LMMs trained using the native visual encoder demonstrate considerable weakness in robustness and foundational visual perception data.

---

> > ### Author Response · Authors · 2024-11-19
> >
> > > **Q 4**:Impact of High-Resolution Inputs
> >
> > **Response:**
> >
> > Thank you for your detailed review.  We pre-trained the model under the configuration of 8XA100 for 10K steps with input images of varying resolutions and summarized the experimental results in the table below. We observed that increasing the input image resolution improves the model's performance and accelerates convergence (with loss < 3). However, this inevitably leads to greater training overhead.
> >
> > | convergence (loss<3) | Resolution | cost         | RVQA-A    | RVQA-R    | RVQA-V    |
> > | :------------------- | :--------- | ------------ | --------- | --------- | --------- |
> > | 2013 steps           | 256        | 9.25  s/step | 52.69     | 60.73     | 59.37     |
> > | 1764 steps           | 448        | 9.74  s/step | **52.94** | **61.52** | **60.86** |
> >
> > > **Q 5**:Diffusion Model's Role
> >
> > **Response:**
> >
> > Thank you for your detailed review.  In order to facilitate a vivid comparison, we have incorporated a dynamic visualization process and made revisions in the draft (highlighted in lines 1067-1071 with red). In Figure 9, we present multiple corrections for the same semantically ambiguous sample, showing that the loss decreases progressively while the semantics are successfully refined.
> >
> > > **Concern 1**: approach is not entirely novel, as seen in works like Transfusion
> >
> > **Response:**
> >
> > Thank you for your detailed review. Although both DEEM and Transfusion utilize diffusion models, their foundational objectives are markedly different. Firstly, the aim of Transfusion is to unify the two architectures of transformers and diffusion models to completely integrate image and text generation and understanding, thereby enhancing the general capabilities of language models. In contrast, DEEM seeks to improve the fundamental perceptual abilities of language models by eliminating the latent misinterpretations of semantic information that arise from training on low-quality image-text pairs with dependent visual encoders. Secondly, in terms of outcomes, Transfusion exhibits significantly poor performance on general visual-language tasks, whereas DEEM not only substantially enhances the foundational visual perception capabilities of language models but also demonstrates commendable performance on general visual-language tasks. This evidence indicates that Transfusion has failed in its unification process, whereas DEEM has achieved success.
> >
> > > **Concern 2**: If a diffusion module is introduced, the authors should also focus on evaluating image generation capabilities.
> >
> > **Response:**
> >
> > There are numerous methods that employ diffusion models for representation learning. While diffusion models are particularly well-suited for generative tasks, their robust generative capabilities make them equally suitable for conditional guidance in representation learning [3,4]. In the related work section, we also present a variety of pertinent studies.
> >
> > [3] Chen Wei, et al."De-Diffusion Makes Text a Strong Cross-Modal Interface." CVPR (2024).
> >
> > [4] Wang, Wenxuan, et al. "Diffusion Feedback Helps CLIP See Better." *arXiv preprint arXiv:2407.20171* (2024).
> >
> > Thank you again for your insightful comments.  If you have other comments, we are happy to address them to polish this work. We look forward to contributing to the development of both the Multi-Modal research and the open-source community.

---

> ### Author Response · Authors · 2024-11-21
>
> Dear Reviewer qAMc:
>
> We greatly appreciate the time and effort you dedicated to reviewing our paper. We have carefully addressed all your insightful suggestions and clarified any ambiguous points to improve our work. As the deadline for the discussion is nearing, could you kindly reconsider your evaluation based on the revised version? We are open to any further queries you might have and are eager to provide any additional information needed.
>
> Thank you for your understanding and support.
>
> Best regards,
>
> Authors

---

> ### Author Response · Authors · 2024-11-22
>
> Dear Reviewer qAMc:
>
> Thanks a lot for your efforts in reviewing this paper. We tried our best to address the mentioned concerns. As the discussion deadline between reviewers and authors is very close, we tend to confirm whether there are unclear explanations and descriptions here. We could further clarify them.
>
> Thanks!
>
> Authors

---

> ### Author Response · Authors · 2024-11-24
>
> Dear Reviewer qAMc,
>
> I hope this email finds you in good spirits. We are grateful for your detailed evaluation and have worked hard to address your concerns. With the discussion deadline fast approaching, we would appreciate it if you could revisit the manuscript to see if the adjustments align with your expectations. Your expertise continues to play a pivotal role in refining our work.
>
> Thank you once again for your dedication.
>
> Sincerely,
>
> Authors

---

> > ### Comment · Reviewer_qAMc · 2024-11-25
> >
> > Sorry for the late reply. Thank the author for the rebuttals. My main concerns have been addressed, so I increase my score to 6. Looking forward for you open-sourced codebase and models.

---

### Official Review · Reviewer_CASC · 2024-10-31

**Soundness:** 3
**Presentation:** 3
**Contribution:** 3
**Rating:** 8
**Confidence:** 5

**Summary:**

The DEEM model integrates diffusion models with large multimodal models (LMMs) to enhance image perception capabilities, especially for out-of-distribution data. DEEM employs generative feedback from diffusion models to correct potential semantic biases in image encoders, resulting in more accurate and robust visual perception. Evaluations across benchmarks such as RobustVQA, POPE, and MMVP demonstrated improvements in visual robustness and reduced visual hallucinations compared to traditional methods. This enhancement is achieved without additional training parameters, using fewer resources and a smaller model scale.

**Strengths:**

1.	Innovation in Multimodal Integration: DEEM introduces diffusion models as “visual auditors” for LMMs, a novel approach for refining visual inputs and reducing errors from biased image encoders.
	2.	Efficiency: The approach reduces dependency on larger, more complex models by using a smaller image encoder (CLIP-ConvNext-B), which lowers computational requirements.
	3.	Robustness: Demonstrates resilience against adversarial and out-of-distribution data, enhancing the model’s suitability for diverse, real-world scenarios.
	4.	Extensive Evaluation: The paper provides a comprehensive analysis using custom (RobustVQA) and widely recognized benchmarks (POPE and MMVP), showing DEEM’s performance gains in both robustness and visual perception.

**Weaknesses:**

1.	Reliance on Diffusion Models: Although diffusion models enhance robustness, they also introduce computational complexity. Future versions might explore alternative approaches that achieve similar robustness with lower computational costs.
	2.	Generalization Limits: While DEEM shows strong performance for specific multimodal tasks, its generalization to other modalities (e.g., audio) remains unexplored.
	3.	Training Overheads: The training process, though optimized, may still be prohibitive for some use cases where computational resources are constrained.

**Questions:**

1. Could the diffusion model’s role be optimized or reduced without compromising robustness? For instance, could a selective application of diffusion-based feedback lower training costs?
2. Could the impact of scaling down the diffusion model be evaluated further to assess how smaller models might perform in balancing robustness and efficiency?
3. Have you considered a comparative analysis of DEEM with approaches like "De-Diffusion Makes Text a Strong Cross-Modal Interface"?

https://openaccess.thecvf.com/content/CVPR2024/papers/Wei_De-Diffusion_Makes_Text_a_Strong_Cross-Modal_Interface_CVPR_2024_paper.pdf

---

> ### Author Response · Authors · 2024-11-19
>
> Thanks for the valuable and encouraging comments! Our point-by-point responses to the reviewer's mentioned concerns are provided as follows.
>
> > **W 1**:  Future versions might explore alternative approaches that achieve similar robustness with lower computational costs.
>
> **Response:**
>
> Thank you for your insightful concern. An important direction for the future is to optimize our model to achieve comparable performance with reduced computational costs, for instance, by shifting from an online learning approach to a suboptimal offline learning strategy.
>
> > **W 2**: While DEEM shows strong performance for specific multimodal tasks, its generalization to other modalities (e.g., audio) remains unexplored.
>
> **Response:**
>
> Thank you for your detailed review. Our approach primarily focuses on visual language models, and exploring modal generalization in other bi-modal models, such as audio language models, represents an important direction for future research.
>
> > **W 3**:The training process, though optimized, may still be prohibitive for some use cases where computational resources are constrained.
>
> **Response:**
>
> Thanks for your professional and careful review. An improvement can be made by replacing online learning with offline learning. By conducting self-supervised training on the visual encoder in the image modality, we can obtain an optimized visual encoder that can be integrated with a large language model (LLM) for training on image-text instruction data. This approach enhances the applicability of our method and allows us to improve model capabilities even in low-resource scenarios.
>
> > **Q 1**:Could the diffusion model’s role be optimized or reduced without compromising robustness? For instance, could a selective application of diffusion-based feedback lower training costs?
>
> **Response:**
>
> Thank you for your detailed review. We believe that it is quite likely that visual encoders do not exhibit semantic bias in all images. If an effective offline data selection method is designed, offline selection of those valid and learnable samples could significantly reduce training costs while maintaining similar performance.
>
> > **Q 2**: Could the impact of scaling down the diffusion model be evaluated further to assess how smaller models might perform in balancing robustness and efficiency?
>
> **Response:**
>
> Thank you for your valuable comments. It is feasible to explore the trade-off between robustness and efficiency. Current diffusion models are categorized into base, medium, and large scales. In this paper, we utilize a base-scale diffusion model with approximately 800M parameters. If we aim to conduct scaling-down experiments, the absence of smaller-scale pre-trained diffusion models necessitates the pre-training of multiple diffusion models with parameters ranging from 40M to 800M, as described in [1]. We will try our best to investigate this topic as a key research direction in the future.
>
> [1] Kangfu Mei, et al. "Bigger is not Always Better: Scaling Properties of Latent Diffusion Models"  *arXiv preprint arXiv:2404.01367* (2024).
>
> > **Q 3**: Have you considered a comparative analysis of DEEM with approaches like "De-Diffusion Makes Text a Strong Cross-Modal Interface"?
>
> **Response:**
>
> Thank you for your detailed review. The focus of DEEM is to utilize diffusion feedback to enable end-to-end learning of continuous image spaces within MLLMs, allowing LLMs to better perceive complex out-of-distribution (OOD) images. In contrast, "De-Diffusion Makes Text a Strong Cross-Modal Interface" aims to train effective image-to-text feature descriptions while employing a discretization approach in its learning process and assessing the general capabilities of LLMs. Due to  time and resources limits, we found it challenging to conduct a direct comparison. Nevertheless, we have referenced and highlighted our modifications in related works to enhance the theoretical support for our research.
>
>
>
> Thank you again for your insightful comments.  If you have other comments, we are happy to address them to polish this work. We look forward to contributing to the development of both the Multi-Modal research and the open-source community.

---

> ### Author Response · Authors · 2024-11-21
>
> Dear Reviewer CASC:
>
> Thank you immensely for your valuable feedback on our manuscript. We've worked diligently to incorporate your suggestions and make necessary revisions. With the review timeline approaching, we kindly ask if you could spare a moment to re-evaluate the updated document. Please let us know if there is anything else you need from our end for clarification.
>
> We truly appreciate your cooperation and continued support.
>
> Warm regards,
>
> Authors

---

> ### Comment · Reviewer_CASC · 2024-11-21
>
> Thank you for the detailed feedback. I've raised my rating.

---

### Official Review · Reviewer_9Tws · 2024-11-03

**Soundness:** 4
**Presentation:** 3
**Contribution:** 4
**Rating:** 8
**Confidence:** 3

**Summary:**

This paper proposed a method to enhance the image perception ability of large multimodal models (LMMs), by leveraging the generative feedback of diffusion models. This reduced the visual hallucination and strengthen the perception ability in against out-of-distribution samples. Also a benchmark is developed, specifically designed for out-of-distribution or challenging samples. Extensive experiments showed that the proposed method achieves an improvement in a collection of benchmarks and downstream tasks.

**Strengths:**

The paper is well-structured and it shows a good originally. This is a pioneering work in using diffusion models to specifically enhance the image perception capabilities of LMMs.

**Weaknesses:**

W1) The detail information of RobustQA is missing. This makes the experiments result look less convincing. How many samples are in the benchmark? Please also include a diversity chart to show how robust the benchmark is.

W2) Figure 2 is difficult to follow. Please revise the figure, e.g. adding some step indicators, legends for arrows.

W3) Typo "obnly" in the last paragraph of Section 3.3.

**Questions:**

Q1) Regarding RobustVQA, would it be possible to also run on close-source models (GPT4o/Gemini1.5Pro)? I am curious on how the close-source solutions perform against the proposed benchmark.

Q2) In Table 9, it states "Questions in the yes or no format can well evaluate the performance of the models on the RobustVQA benchmark, while questions in the multiple choice format are very random, and MM-interleaved tend to output the first option. Therefore, we
adopt yes or no format in our experimental settings." seems a bit unconvincing to me. Would LMM also be easier to guess the solution if there is only 2 choices for yes/no format?

---

> ### Author Response · Authors · 2024-11-19
>
> Thanks for the valuable and encouraging comments! Our point-by-point responses to the reviewer's mentioned concerns are provided as follows.
>
> > **W 1**: The detail information of RobustQA is missing.
>
> **Response:**
>
> Thank you for your detailed review. We details the construction process in appendix. We filter easy-classification samples using CLIP-VIT-L and then construct positive-negative pairs by using the ground truth and the highest confidence incorrect category with a Yes/No question template like POPE. For each image sample, only two categories are used. The sample numbers and class categories comparison are as follows:
>
> | Dataset    | ImageNet-A | ImageNet-R | ImageNet-V2 | RVQA-A | RVQA-R | RVQA-V |
> | ---------- | ---------- | ---------- | ----------- | ------ | ------ | ------ |
> | samples    | 3725       | 15000      | 2500        | 2356   | 6562   | 2415   |
> | categories | 200        | 200        | 1000        | 199    | 200    | 997    |
>
> As shown in the table, the quantity and diversity of the RobustVQA are well ensured. Moreover, our RobustVQA can be combined with other benchmarks such as POPE and MMVP to provide a more comprehensive assessment of model robustness.
>
> > **W 2**:Figure 2 is difficult to follow. Please revise the figure, e.g. adding some step indicators, legends for arrows
>
> **Response:**
>
> Thank you for your valuable review. We have made the relevant modifications in the manuscript and highlighted them in red, including the addition of step indicators and legends for the arrows, in order to enhance clarity. Thank you for your valuable suggestions, which have provided us with the opportunity to polish our work.
>
> > **W 3**:Typo "obnly" in the last paragraph of Section 3.3.
>
> **Response:**
>
> Thank you for your insightful review. We have made the relevant modifications in the manuscript and highlighted them in red.
>
> > **Q 1**:it be possible to also run on close-source models (GPT4o/Gemini1.5Pro).
>
> **Response:**
>
> Sure!. Due to time limits, we present the results of GPT4o and GPT4o-mini on RobustVQA as below.
>
> | Architecture | RVQA-A | RVQA-R | RVQA-V |
> | ------------ | ------ | ------ | ------ |
> | GPT4o        | 57.42  | 65.86  | 55.95  |
> | GPT4o-mini   | 52.94  | 57.22  | 50.17  |
>
> > **Q 2**:Would LMM also be easier to guess the solution if there is only 2 choices for yes/no format?
>
> **Response:**
>
> Thanks for your professional and careful review. The selection bias varies across different LLMs[1], and the presence of selection bias can lead to unfair comparisons in multiple-choice questions. Furthermore, as the number of options increases, the cost associated with ensuring a fair comparison rises significantly. Therefore, utilizing a yes/no format allows for fair comparisons to be made with minimal cost, thereby enabling a stable and efficient assessment of the performance differences among various models.
>
> [1] Chujie Zheng, et al. "Large Language Models Are Not Robust Multiple Choice Selector"  ICLR 2024
>
>
>
> Thank you again for your insightful comments.  If you have other comments, we are happy to address them to polish this work. We look forward to contributing to the development of both the Multi-Modal research and the open-source community.

---

> ### Author Response · Authors · 2024-11-21
>
> Dear Reviewer 9Tws:
>
> We are truly grateful for your insightful comments and the guidance you provided during your review of our paper. We are pleased to inform you that we have addressed all points raised and have made significant improvements. As the discussion phase draws near, we kindly request your reevaluation at your earliest convenience. Should any questions remain, we are at your disposal to clarify them promptly.
>
> Thank you for your time and understanding.
>
> Sincerely,
>
> Authors

---

> > ### Comment · Reviewer_9Tws · 2024-11-21
> > **Response to Rebuttal.**
> >
> > Thanks for the detailed response, I have adjusted my score (6 -> 8).

---

### Official Review · Reviewer_ikTv · 2024-11-03

**Soundness:** 3
**Presentation:** 3
**Contribution:** 3
**Rating:** 8
**Confidence:** 3

**Summary:**

This paper enhances large language models image perception by using diffusion models to align image encoder semantics, improving resilience to out-of-distribution data and reducing visual hallucinations. Without extra modules or parameters, DEEM outperforms on benchmarks like RobustVQA, POPE, and MMVP, using fewer parameters and less data. It strengthens visual tasks such as question answering, captioning, and image synthesis, showcasing robust perception and reduced hallucinations.

**Strengths:**

1. This paper design a new robustness benchmark RobustVQA for visual robustness capabilities of multi modal models.
2. DEEM is the first approach to enhance image perception in LLMs using diffusion models, making it novel.
3. This approach is highly versatile, demonstrating effective experimental results across benchmarks.
4. This paper is well written and easy to follow.

**Weaknesses:**

1. Since diffusion models need to be utilized during both training and inference, the computational cost will be high.
2. If this model is used with an LLM that has a much larger number of parameters, how would its performance be affected?

**Questions:**

Please refer to weaknesses.

---

> ### Author Response · Authors · 2024-11-19
>
> Thanks for the valuable and encouraging comments! Our point-by-point responses to the reviewer's mentioned concerns are provided as follows.
>
> > **W 1**: Since diffusion models need to be utilized during both training and inference, the computational cost will be high.
>
> **Response:**
>
> Thank you for your insightful concern. There may be some misunderstandings that we would like to clarify further:
>
> 1. The additional computational overhead introduced by the diffusion model is not related to the size of large language models or visual encoders. it is solely dependent on the size of the diffusion model.
> 2. During the application of the diffusion model, model parameters are frozen, allowing for manageable training costs. Furthermore, numerous acceleration techniques [1] can be employed to optimize inference speed.
>
> [1] Haozhe Liu, et al. "Faster Diffusion via Temporal Attention Decomposition" NeurIPS2024
>
> > **W 2**: If this model is used with an LLM that has a much larger number of parameters, how would its performance be affected?
>
> **Response:**
>
> Thank you for your detailed review. We explored the impact of scaling up the model size of LLM on the approach and conducted ablation experiments, with the results presented in the table below. We found that DEEM exhibits scalability, meaning that increasing the parameters of the LLMl can still lead to performance improvements.
>
> | Architecture          | training  data | RVQA-A | RVQA-R | RVQA-V | POPE-R | POPE-P | POPE-A | OK-VQA |
> | --------------------- | -------------- | ------ | ------ | ------ | ------ | ------ | ------ | ------ |
> | ConvNext-L/Vicuna 7B  | 160k           | 53.23  | 60.47  | 56.88  | 61.12  | 62.87  | 62.09  | 23.87  |
> | ConvNext-B/Vicuna 13B | 160k           | 53.92  | 61.27  | 57.02  | 62.60  | 64.26  | 63.19  | 31.11  |
>
>
>
> Thank you again for your insightful comments.  If you have other comments, we are happy to address them to polish this work. We look forward to contributing to the development of both the Multi-Modal research and the open-source community.

---

> ### Author Response · Authors · 2024-11-21
>
> Dear Reviewer ikTv:
>
> We hope this message finds you well. We deeply appreciate your thoughtful feedback and the attention you’ve given to our manuscript. All concerns have been thoroughly addressed, and we wish to invite you to review the manuscript once more. With the deadline approaching, we would be grateful if you could confirm that all uncertainties have been resolved. We are ready to assist you with any further clarifications.
>
> Thanks for your cooperation.
>
> Kind regards,
>
> Authors

---

> > ### Comment · Reviewer_ikTv · 2024-11-22
> >
> > Thank you for your response. I checked the reviews of other reviewers. I believe this paper will have a positive impact on the research area. I will maintain my score as is. Thanks.

---

### Official Review · Reviewer_Sncp · 2024-11-06

**Soundness:** 3
**Presentation:** 3
**Contribution:** 3
**Rating:** 6
**Confidence:** 4

**Summary:**

This paper introduces the diffusion model into the image perception process of large language models to reduce excessive compression of visual details, thereby enhancing the model's robustness and its ability to mitigate hallucinations. An end-to-end multimodal model with interleaved text-image modeling capabilities is trained to perform various multimodal tasks in a unified manner. A new robustness benchmark for large multimodal models is developed to assess their visual robustness capabilities. The extensive experimental results demonstrate the effectiveness of the proposed method.

**Strengths:**

1. The paper introduces a new benchmark, RobustVQA, specifically designed for evaluating the visual robustness capabilities of the large multimodal models.
2. The proposed method is the first to introduce the diffusion model into the image perception of large language models, which is simple but effective.
3. The proposed method trained end-to-end with interleaved text-image data can perform various multimodal tasks in a unified manner.

**Weaknesses:**

1. The insight of the paper is the integration of the diffusion model into the image perception process of LLMs while training to correct potential semantic biases in the image encoder. It is crucial to validate the generalization capability of the proposed method across existing LMMs to enhance their image perception. However, the validation is somewhat limited, focusing solely on LLaVA, as shown in Table 5. Could the proposed method be flexibly applied to additional existing architectures, such as BLIP-2 and MiniGPT-4?
2. The claim that the proposed method can "correct potential semantic bias in the image encoder and alleviate excessive compression of visual details" seems intuitive. Are there any validations or qualitative comparisons between existing LMMs trained with and without this method? For instance, maybe comparisons of feature maps for image tokens encoded by the image encoder trained with and without the proposed method.
3. The contribution of the constructed RobustVQA benchmark seems somewhat limited, as it focuses primarily on evaluating visual robustness. Emphasizing the unique advantages of the RobustVQA benchmark over existing benchmarks through more extensive qualitative or quantitative comparisons with other benchmarks in some aspects would enhance its value.

**Questions:**

See Weaknesses

---

> ### Author Response · Authors · 2024-11-19
>
> Thanks for your professional and careful review. We respond to your concerns or questions as follows.
>
> > W 1: Could the proposed method be flexibly applied to additional existing architectures, such as BLIP-2 and MiniGPT-4?
>
> **Response:**
>
> Thank you for your insightful concern. DEEM can be flexibly applied to connect visual encoders with large language models using a bridging network in text generation-only LMMs, such as LLaVA, BLIP2, and MiniGPT-4. Our connector, the Perceiver Resampler [1], can dynamically transform visual embeddings of arbitrary lengths into the desired length and dimensionality, which can then be fed into a diffusion model.
>
> [1] Alayrac, Jean-Baptiste, et al. "Flamingo: a visual language model for few-shot learning." NeurIPS2022
>
> > W 2: Are there any validations or qualitative comparisons between existing LMMs trained with and without this method?
>
> **Response:**
>
> Thanks for your valuable comments. In order to facilitate a vivid comparison, we have incorporated a dynamic visualization process and made revisions in the draft (highlighted in lines 1067-1071 with red). In Figure 9, we present multiple corrections for the same semantically ambiguous sample, showing that the loss decreases progressively while the semantics are successfully refined.
>
> > W 3: more extensive qualitative or quantitative comparisons with other benchmarks in some aspects would enhance its value.
>
> **Response:**
>
> We details the construction process in appendix. Specifically, we filter easy-classification samples using CLIP-VIT-L and then construct positive-negative pairs by using the ground truth and the highest confidence incorrect category with a Yes/No question template like POPE. For each image sample, only two categories are used. The sample numbers and class categories comparison are as follows:
>
> | Dataset    | ImageNet-A | ImageNet-R | ImageNet-V2 | RVQA-A | RVQA-R | RVQA-V |
> | ---------- | ---------- | ---------- | ----------- | ------ | ------ | ------ |
> | samples    | 3725       | 15000      | 2500        | 2356   | 6562   | 2415   |
> | categories | 200        | 200        | 1000        | 199    | 200    | 997    |
>
> As shown in the table, the quantity and diversity of the RobustVQA are well ensured. Moreover, our RobustVQA can be combined with other benchmarks such as POPE and MMVP to provide a more comprehensive assessment of model robustness.
>
>
>
> Thank you again for your insightful comments.  If you have other comments, we are happy to address them to polish this work. We look forward to contributing to the development of both the Multi-Modal research and the open-source community.

---

> ### Author Response · Authors · 2024-11-21
>
> Dear Reviewer Sncp:
>
> Thank you for your extensive review and constructive comments on our manuscript. We have earnestly worked on resolving all issues highlighted and have provided comprehensive responses to your queries. As the review deadline is imminent, we humbly request your reevaluation of our revised submission. Please feel free to reach out if any further clarification is required from our side.
>
> Your support is highly valued, and we thank you in advance for your consideration.
>
> With gratitude,
>
> Authors

---

> ### Author Response · Authors · 2024-11-22
>
> Dear Reviewer Sncp:
>
> We sincerely appreciate your efforts in reviewing this paper! We tried our best to address the mentioned concerns about related work, experimental results, and more empirical evidence.
> The discussion deadline is approaching now. Are there unclear explanations and descriptions here?
> We are highly encouraged if your concerns have been addressed. On the contrary, if you need any more clarification, we can provide it as soon as possible before the discussion deadline.
>
> Thanks!
>
> Authors

---

> > ### Author Response · Authors · 2024-11-24
> >
> > Dear Reviewer Sncp,
> >
> > Thank you very much for your valuable feedback on our paper. We greatly appreciate the time and effort you have invested in reviewing our work. We have carefully considered your comments and have made revisions to address the concerns you raised.
> >
> > As the discussion deadline is approaching, we want to ensure that all explanations and descriptions in our responses are clear and satisfactory. If there are any areas that still seem unclear or require further clarification, please let us know, and we will be happy to provide additional information or make necessary adjustments.
> >
> > We look forward to your feedback and hope that our revised manuscript meets your expectations.
> >
> > Thank you once again for your insightful suggestions.
> >
> > Sincerely,
> >
> > Author

---

> > > ### Comment · Reviewer_Sncp · 2024-11-24
> > >
> > > Thank you for providing additional details and clarifications in your response.
> > >
> > > 1. Flexibility: I believe that DEEM can be flexibly applied to facilitate the visual perception of existing LMMs. From my point of view, including quantitative results of BLIP-2 and MiniGPT-4 with DEEM in Table 5, if feasible, could further demonstrate its value to the MLLM community for reducing visual hallucinations of the image encoder.
> > > 2. Validations: Thank you for providing additional validations of DEEM's effectiveness. I think this essential validation is crucial for inclusion in the main manuscript to verify that DEEM can correct potential semantic biases in the image encoder of LLMs.
> > > 3. Benchmarks: Thank you for providing additional quantitative details of RobustVQA. From my perspective, including a quantity and diversity chart to show the unique advantages of RobustVQA would further enhance its value to the MLLM community.
> > >
> > > Thanks once again for your time!

---

> ### Author Response · Authors · 2024-11-25
>
> Dear Reviewer Sncp,
>
> We hope this message finds you well. We are writing to sincerely thank you for taking the time to review our rebuttal and for providing such insightful and constructive feedback on our submission. Your expertise and attention to detail are truly appreciated and have been incredibly beneficial in guiding our research.
>
> The suggestions you provided have opened up new perspectives and opportunities for enhancing the quality of our work. We are committed to implementing your recommendations to the best of our ability and are eager to explore the ideas you have mentioned further.
>
> Thank you once again for your invaluable contribution to our research. Your support and guidance are deeply appreciated, and we are hopeful that the revisions will meet your expectations.
>
> Sincerely,
>
> Author

---

> > ### Comment · Reviewer_Sncp · 2024-11-30
> >
> > Thanks for the detailed response. I would like to maintain my rating to marginally above the acceptance threshold.

---

### Author Response · Authors · 2024-11-19

We appreciate the reviewers insightful comments and constructive feedback on our manuscript. We are pleased to receive positive ratings from most of the reviewers. Furthermore, we are delighted to learn that the reviewers found the research problem to be significant and the core idea to be novel (Reviewer Sncp, ikTv, 9Tws, CASC, and qAMc), and the experiments to be convincing (Reviewer Sncp, ikTv, 9Tws, CASC and qAMc). Based on the reviews, we provide a general response to the points raised by multiple reviewers and individual responses below to address each reviewer's concerns.

(1) Regarding the questions about the experiments, we have taken the following actions:

- ﻿﻿For Reviewers Sncp, ikTv, 9Tws, and qAMc, we have either highlighted the location of the required experiments corresponding to their comments in our paper or added the pertinent experiments accordingly.
- ﻿﻿For Reviewers Sncp and 9Tws, we have provided a more detailed distribution and construction of the data related to RobustVQA.
- ﻿﻿For Reviewer ikTv, we have conducted ablation experiments to assess the impact of scaling the language model on its performance.
- ﻿﻿For Reviewer 9Tws, we have included comparative experiments of commercial models on the RobustVQA dataset.
- ﻿﻿For Reviewer CASC, due to the constraints of computational resources and the time allocated for rebuttal, we will try our best to validate our method by exploring the trade-off between robustness and efficiency through the downscaling of the diffusion model.
- ﻿﻿For Reviewer qAMc, we have presented experiments examining the effects of increasing input image resolution on model convergence, performance, and additional training overhead.

(2) We have addressed the questions about the idea and technical details as follows:

- ﻿﻿For Reviewers Sncp and qAMc, we have included additional visualizations of the core dynamic processes involved in semantic bias elimination, providing an intuitive representation of the fundamental ideas of our approach to assist the readers in better understanding the mechanisms at play.
- ﻿﻿For Reviewer Sncp, we illustrated the technical details regarding the extraction of visual features and their integration into the diffusion model. This portion of the framework is architecture-agnostic and can be flexibly applied to various latent variable models.
- ﻿﻿For Reviewers ikTv and CASC, we further clarified the optimization strategies and methodologies available to mitigate the additional computational overhead introduced by diffusion feedback, enabling our model to be applied more effectively without constraints.
- ﻿﻿For Reviewer 9Tws, we discussed the limitations of the multiple-choice format and provided a rationale for employing the yes/no format instead.
- ﻿﻿For Reviewer 9Tws, we corrected typographical errors and revised our Figure 2 in accordance with their feedback to enhance its clarity.
- ﻿﻿For Reviewer qAMc, we further explained the differences between DEEM and Transfusion.
- ﻿﻿For Reviewer qAMc, we explored the reasons behind Robustness Knowledge Forgetting and proposed potential solutions.
- ﻿﻿For Reviewer qAMc, we further clarified the misconceptions regarding the case study, explaining that it involves the visualization of randomly selected images, which includes both successful and unsuccessful cases.

(3) Missing reference:

- ﻿﻿For reviewer CASC, we have included a detailed discussion about "De-Diffusion Makes Text a Strong Cross-Modal Interface" in the related work section in the revised draft.

We also have revised the draft according to all the reviewers. The revisions are highlighted in red. We sincerely thank all the reviewers for their constructive suggestions. Please feel free to let us know if further details/explanations would be helpful.

Yours truly,

Authors of #5298

---

### Meta-Review · Area_Chair_hzcT · 2024-12-21

**Metareview:**

This paper has received ratings of 8, 8, 8, 6, 6, where the reviewers have generally rated the paper positively, highlighting its innovative approach to enhancing visual robustness and mitigating hallucinations in multimodal models.

The authors has proposed a framework DEEM, leveraging diffusion models to enhance the visual perception of large multimodal models (LMMs). The authors demonstrate that integrating generative feedback from diffusion models can address limitations of traditional image encoders, improving performance on tasks such as visual question answering, image captioning, and text-conditioned image synthesis.

Strength:
- Innovation in Methodology: The proposal to use diffusion models as a corrective mechanism for image encoders is novel and effective.
- Empirical Results are also promising. The model achieves state-of-the-art performance on several benchmarks, including RobustVQA, POPE, and MMVP, with notable improvements in robustness and hallucination mitigation.
- Scalability and Efficiency: The method demonstrates competitive performance with fewer parameters and less pre-training data compared to contemporary models.

Area for improvements:
- The computational requirements for integrating diffusion models may limit practical applications.
- The broader applicability of the approach to larger image encoders and other multimodal tasks requires further exploration.

The manuscript makes a significant contribution to the field of multimodal modeling by advancing the use of diffusion models in enhancing image perception. The empirical results are strong, and the methodology is robust and generalizable. While the computational overhead and scalability concerns are valid, they do not outweigh the novelty and impact of the work.

**Additional Comments On Reviewer Discussion:**

Authors and reviewers were actively engaged, leading to a smooth, constructive, and productive discussion.

The reviewers initially raised concerns about the novelty of leveraging diffusion models within large multimodal frameworks and the clarity of the methodology section. They also highlighted the need for more detailed comparisons to similar state-of-the-art models and additional ablation studies to substantiate the proposed method's contributions. In their rebuttal, the authors effectively clarified their contribution of integrating diffusion models as a corrective mechanism for visual robustness and hallucination mitigation in multimodal models. The authors further expanded their explanations of the technical approaches, addressed specific methodological concerns, and provided extended experimental results, including detailed ablation studies and comparisons against competing methods.

The reviewers appreciated these efforts, noting that the additional details provided in the rebuttal and the newly presented results significantly improved the paper's overall rigor and presentation. As a result, the review process concluded with a consensus on the strengths of the proposed method and its potential contributions to the field.

---

### Decision · Program_Chairs · 2025-01-22

Accept (Spotlight)